

# Observing system simulation experiments reveal that subsurface temperature observations improve estimates of circulation and heat content in a dynamic western boundary current

David E. Gwyther[1], Colette Kerry[1], Moninya Roughan[1], and Shane R. Keating[2]

[1]Coastal and Regional Oceanography Lab, School of Biological, Earth and Environmental Sciences, UNSW Sydney, Sydney, NSW, Australia
[2]School of Mathematics and Statistics, UNSW Sydney, Sydney, NSW, Australia

**Correspondence:** David E. Gwyther (david.gwyther@gmail.com)

**Abstract.** Western boundary currents (WBCs), form the narrow, fast-flowing poleward return flows of the great subtropical ocean gyres and are sources of rapidly varying mesoscale eddies. Accurate simulation of the vertical structure, separation latitude, and ocean heat content of WBCs is important for understanding the poleward transport of heat in the global ocean. However, state estimation and forecasting in WBC regions, such as the East Australian Current (EAC), the WBC of the South

Pacific subtropical gyre, is challenging due to their dynamic nature and lack of observations at depth. Here we use Observing System Simulation Experiments to show that subsurface temperature observations in a high eddy kinetic energy region yield large improvement in representation of key EAC circulation features, both downstream and ∼600 km upstream of the observing location. These subsurface temperature observations (in concert with sea surface temperature and height measurements) are also critical for correctly representing ocean heat content along the length of the EAC. Furthermore, we find that a more

poleward separation latitude leads to an EAC and eddy field that is represented with far reduced error, compared to when the EAC separates closer to the equator. Our results demonstrate the importance of subsurface observations for accurate state estimation of the EAC and ocean heat content that can lead to marine heatwaves. These results provide useful suggestions for observing system design under different oceanographic regimes, for example, adaptive sampling to target high energy states with more observations and low energy states with fewer observations.

## 1 Introduction

Western Boundary Currents (WBCs) transport warm and saline waters towards the poles, and are key regions for eddy generation. Hence, they play a critical role in weather and climate, ecosystems and biogeochemistry. They contribute to cross shelf exchange with their adjacent coastal seas (Brink, 2016) and hence influence local Blue Economies (e.g. Li et al., 2017; Zeng et al., 2018). Given the proximity in location to populated coastlines and the dominant role in coastal ocean processes,

characterising and predicting WBCs and their related eddy fields is a subject of intensive observing and modelling efforts.

Due to the chaotic nature of mesoscale circulation, particularly WBCs, ocean models must be regularly updated through the incorporation of observations in order to correctly represent rapidly changing ocean conditions. Data Assimilation (DA)





combines observations and a numerical model in a dynamically consistent way such that the result is a better estimate of the ocean circulation than either alone (Moore et al., 2019). Due to the expense of observational oceanography and the vast nature of the ocean, there is strong motivation to optimise the results that are obtained from assimilation of sparse observations and to provide insight into both ocean dynamics and guidance for designing optimised observing systems.

Observing System Simulation Experiments (OSSEs) provide a means by which the impact of assimilating different observations can be assessed using synthetic observations. In an OSSE, a model simulation is taken as representing the 'true' state of the ocean that (unlike the real ocean) is completely known without error (Halliwell et al., 2014). Values are extracted from this simulation and realistic errors are added to represent synthetic observations. The impact of assimilating these synthetic observations can then be quantified by comparing the 'truth' and the results of the OSSE forecast and analysis. A thorough examination of the impact of existing and hypothetical observing strategies on key dynamics of interest can then be conducted.

A key advantage of OSSEs is that they allow an assessment of future observational platforms and strategies and so have been used for planning observational experimental design in many applications, for example: an Argo float array (Schiller et al., 2004) and moored instrument array in the Indian Ocean (Oke and Schiller, 2007; Ballabrera-Poy et al., 2007); glider deployments in the Western Atlantic (Halliwell et al., 2017) and the Solomon Sea (Melet et al., 2012); and Argo floats, drifting floats and mooring arrays in the Atlantic (Gasparin et al., 2019). Kamenkovich et al. (2017) used OSSEs to determine the optimal number of autonomous floats to improve representation of biogeochemical variables in the upper Southern Ocean. Using an OSSE framework developed by Halliwell et al. (2014), various observing platforms in the Gulf of Mexico were investigated for improving hurricane prediction and oil spill response. Halliwell et al. (2015) found that deeper profiling XCTDs better constrain upper ocean density (via observations of salinity) and hence ocean pressure and velocity. OSSEs have also been used to examine the impact of existing observation networks, for example, Lee et al. (2020) which showed the relative effect of two serial CTD transects on representing flow patterns of the Kuroshio Current. OSSEs can also reveal the performance of the data assimilation system, such as in Moore et al. (2020), which showed the importance of choice of the assimilation window length.

In this study, we assess the relative impact of various observational platforms on the representation of the East Australian Current through an OSSE framework. The East Australian Current (EAC), like other WBCs, flows adjacent to the continental shelf, has strong currents, and is characterised by the poleward transport of warm and salty tropical waters. However, unlike other WBCs, as the EAC flows offshore, it does not form an extended inertial jet with pseudo-regular formation of eddies from the meandering current as in the Kuroshio Current (Kawabe, 1995) or the Gulf Stream (Richardson and Knauss, 1971). Instead, the EAC can be described as a 'shelf-following' jet, which separates from the continental shelf to feed a poleward and eastward flowing eddy field of anti-cyclonic (counter-clockwise rotation with a warm core) and cyclonic (clockwise rotation with a cold core) eddies (Fig. 1). The latitude of separation varies northward and southward of the mean, typically between 31–32° S (Cetina-Heredia et al., 2014), and has been shown to be linked to the mean kinetic energy of the EAC jet upstream (Li et al., 2022a). The location of the EAC jet, along the continental shelf break from approximately 25° S (north of Brisbane) to 30° S–32.5° S (Coffs Harbour to north of Newcastle), and the subsequent eddy field which stretches south to Tasmania and eastward towards New Zealand are shown in Fig. 1. Key outstanding questions about the EAC focus on the observed warming





of the EAC (Malan et al., 2021), the more frequent and intense nature of Marine Heat Waves (MHWs) in the EAC (Oliver et al.,
2018) and their subsurface structure (Elzahaby and Schaeffer, 2019; Elzahaby et al., 2021), and changes to upper ocean heat
60    content in the EAC (e.g. Li et al., 2022a). These questions are also relevant for analogous WBC systems that have been shown
to be warming 2–3 times the global average (Wu et al., 2012). As well as understanding long term change, there is a strong
desire to improve short term prediction, which is important for extreme weather events, search and rescue, oil spill response
and navigation. Future observing systems will have to be designed to target these key WBC uncertainties but in a cost-effective
manner.

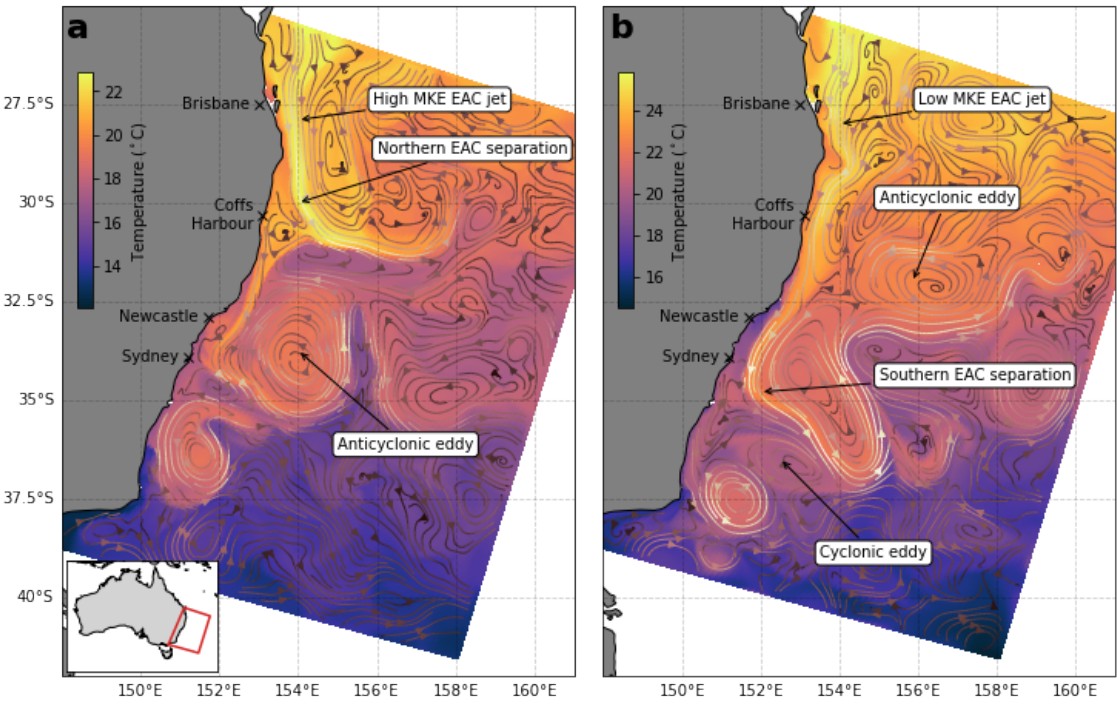

**Figure 1.** Flow regimes of the EAC, highlighted in snapshots of sea surface temperature (colour) and circulation (arrows). The EAC flows
southwards past Brisbane, generally as a coherent jet, before separating from the coast between Sydney and Coffs Harbour. (a) A higher
mean kinetic energy of the EAC jet generally leads to more northern EAC separation, while (b) a lower mean kinetic energy in the EAC
jet generally leads to more southern EAC separation (Li et al., 2021). Flow instability leads to the generation of cyclonic and anticyclonic
eddies. The background is coloured for model sea surface temperature and the streamlines indicate the concurrent surface circulation. Inset
in (a) shows model domain in red outline.

65    Compared to other WBCs, the EAC is relatively well-observed (Roughan et al., 2015; Todd et al., 2019), however, these ob-
servations are expensive and require significant person-hours to obtain. Yet, optimising the impact of different observations on
the EAC has only recently seen dedicated research effort. Kerry et al. (2018) used DA methods to highlight that observations in
regions of high natural variability contribute the most to constraining model solution, while Siripatana et al. (2020) showed the



strong positive impact that high frequency radar and subsurface observations had on improving representation of the subsurface
structure of the EAC. In this study, we employ a time-dependent, variational DA scheme to a shelf-resolving model of the EAC
system and assess how various synthetic datastreams impact estimates of key EAC features. In particular, we examine the role
of surface and subsurface temperature observations in improving the simulation of prominent EAC flow features, the vertical
and spatial heat and velocity distributions, and ocean heat content. The impact of data from a range of observation platforms
on correctly estimating the ocean state are also examined within different EAC separation regimes. In Sect. 2, the model setup
and framework of the OSSEs are introduced while Sect. 3 compares how each OSSE performs across a series of key metrics.
Sect. 4 discusses the impact of assimilating observation on the simulation of EAC dynamics, the representation of ocean heat
content, the influence of EAC separation latitude on state estimates (for example, representation of circulation and velocity),
before finishing with recommendations for future observing system design for this dynamic WBC.

## 2 Methods

### 2.1 Data Assimilating Model

This study uses a numerical ocean model configuration of the Regional Ocean Modeling System (ROMS) and employs the 4-
Dimensional Data Assimilation (4DVar) framework, using an updated setup of Kerry et al. (2016). ROMS is a finite-difference
method ocean model that solves the primitive equations on a horizontal grid with a terrain-following vertical coordinate.
The model domain, with bathymetry sourced from the Geoscience Australia 50 m multibeam survey (Whiteway et al., 2009),
extends from 27° S–38° S, over ∼ 700 km offshore, and is rotated by 20 degrees so as to approximately align the grid parallel to
the coastline (Fig. 1). Horizontal resolution is 2.5 km to 6 km , with the 30 vertical s-coordinate layers with increased resolution
in the surface and bottom boundary layers.

Lateral forcing conditions are taken from BRAN2020 (Chamberlain et al., 2021), comprising currents, temperature and
salinity. Surface forcing conditions are enforced with a bulk flux formulation (Fairall et al., 1996) using daily atmospheric fields
from the Australian Bureau of Meteorology's ACCESS reanalysis (Puri et al., 2013). This model has been used previously for
a number of studies, and further details on validation, forcing and nudging conditions given in Kerry et al. (2016), Li et al.
(2022a) and Li et al. (2021).

The DA scheme is the same as used in Kerry et al. (2016) and Kerry et al. (2018), namely an Incremental Strong Constraint
4-Dimensional Variational scheme (IS4D-VAR; e.g. Moore et al., 2011). This scheme increments the model initial conditions,
boundary conditions and surface forcing such that the difference between the model solution and observations is minimised,
in a least-squares sense, while considering errors in both. This minimisation is performed over an assimilation window (in this
case 5 days) and, given the time-dependent nature of the technique, observation impact can be far-reaching, both upstream and
downstream, as well as forward and backward in time (e.g. Kerry et al., 2018).

The simulations herein are performed over the period November 2011–January 2013. This period is chosen as a repre-
sentative period for several key metrics (see Sect. 2.2) and also coincides with other studies (Kerry et al., 2016). All OSSE
simulations are conducted with the model setup described here.





## 2.2 Assessment of the free-running simulation

A free-running (non-DA) simulation is required for comparison against (as the reference state) and to provide synthetic observations within the OSSE framework. We use a free-running simulation for the same period of time as the OSSEs, and the model
configuration is identical to that used to produce longer term simulations of the EAC system. These longer term simulations have been demonstrated to produce an accurate representation of the mean and variability of the EAC circulation (Kerry and Roughan, 2020), which we further demonstrate with the mean kinetic energy (MKE) and eddy kinetic energy (EKE). The distribution of MKE and EKE (see Sect. 2.5) in the free-running model configuration is robust compared to satellite-derived estimates from AVISO (Archiving, Validation and Interpretation of Satellite Oceanographic Data) altimetry (Kerry et al., 2016;
Li et al., 2021). A variety of similar configurations have been used in previous studies to examine sub-mesoscale circulation (Kerry et al., 2020), EAC seasonal variability (Kerry and Roughan, 2020), interannual variability and energy conversion (Li et al., 2021), eddy-shelf interactions (Malan et al., 2021) and drivers of change in ocean heat content in the EAC southern extension (Li et al., 2022a). Additionally the aforementioned EAC model provides boundary conditions for high resolution nested studies (e.g. Ribbat et al., 2020; Roughan et al., 2022; Li et al., 2022b).

The free-running simulation uses surface forcing from the Bureau of Meteorology Atmospheric high-resolution Regional Reanalysis for Australia (BARRA-R; Su et al., 2019) and lateral boundary forcing from BRAN2020 (see above). Further details of the free-running simulation configuration are given in Kerry and Roughan (2020) and Li et al. (2021). We refer to this 1-year free running simulation as the 'Reference state'.

Li et al. (2021), using a near identical configuration, showed that the sea surface MKE field matches well with AVISO
satellite-inferred sea surface height observations for their long running 1994–2016 simulation - lending confidence to how our model configuration is able to represent real conditions. The chosen 12-month period of the Reference state also displays oceanographic conditions that match the long term mean values (e.g. Li et al., 2021), as shown in comparisons of spatial mean EKE and volume transport between the Reference state and the mean values of these same quantities from the long-running simulation (Fig. A1). The Reference state volume transport varies between 0 and 61 Sv (with a mean of $18 \pm 15$ Sv southwards),
which compares favourably to the long-term (1994–2016) mean and standard deviation in volume transport of $21 \pm 14$ Sv (see Fig. A1a). Likewise, the time-mean EKE of the whole domain for the Reference state ($0.12 \pm 0.027 \mathrm{~m^2\,s^{-2}}$ ) matches well to the long-term mean and standard deviation in EKE of $0.12 \pm 0.028 \mathrm{~m^2\,s^{-2}}$ (see Fig. A1b). Together, these metrics, which represent the upstream EAC and downstream eddy field, show that the time period chosen is representative of the conditions in the EAC over the recent decades.

## 2.3 The OSSE Framework

A typical OSSE is conducted by comparing a free-running and data-constrained simulation, where the data constraints are values taken from the free-running simulation. We demonstrate our OSSE framework in the schematic shown in Fig. 2. The first step is simulating a free-running experiment for a set period of time (the Reference state; Step 1 in Fig. 2). A separate experiment, forced identically to the Reference state but with perturbed initial conditions (see details of perturbation below),



can also be simulated (baseline simulation; Step 1 in Fig. 2). By comparing the baseline simulation to the Reference state, the non-convergence of the initial conditions can be assessed. If the initial conditions provide a sufficient perturbation, then this will be used to perturb the start of each OSSE. Ocean conditions can be extracted from the Reference state to form the synthetic observations (Step 2 in Fig. 2), and the forecast is generated for the $i$th cycle (Step 3; Fig. 2). A reanalysis (with the perturbed initial conditions) is generated through assimilation of the synthetic observations into the forecast cycle, such that

differences between the synthetic observations and the free-running forecast are minimised (Step 4; Fig. 2). The reanalysis is used to initialise the next forecast cycle (Steps 3-4 in Fig. 2). The Reference state and the Reanalysis simulation including the synthetic observations can now be compared (Step 5; Fig. 2).

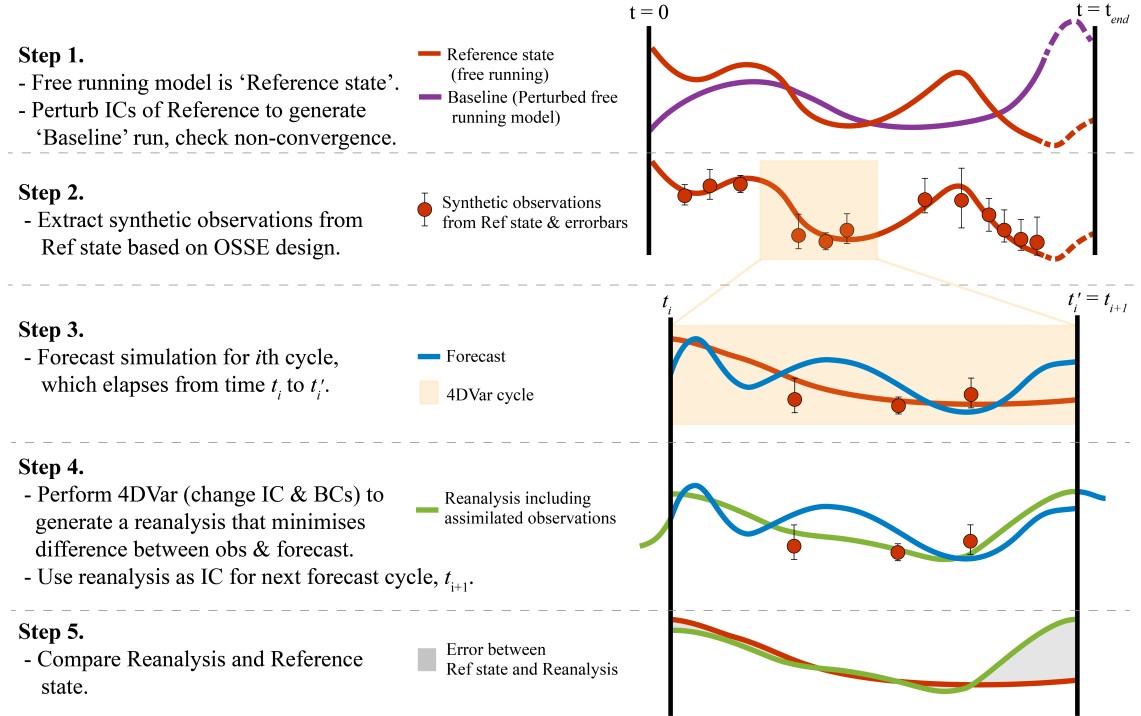

**Figure 2.** The steps taken for conducting and assessing an OSSE using 4DVar are shown schematically.

We initialised each OSSE with initial conditions that were 8-days offset from those that were used to initialise the Reference state. This offset is selected as a balance between an ocean state resulting from the perturbation that is too similar to the unperturbed ocean, such that the DA would not be sufficiently tested, and vastly different conditions, that the DA would

struggle to converge. These perturbed initial conditions were chosen as the perturbation length was slightly less than the calculated de-correlation timescale of volume transport from the Reference state at 28° S (calculated to be ∼9 days). As such by perturbing the initial conditions we aim to represent errors in prediction of the slowly evolving mesoscale ocean circulation, introduced in initial conditions.





We tested a variety of perturbations (1-day, 8-days, 1-month, the climatological mean of November, the initiation month, and 1-year) and calculated the differences in several key metrics (volume transport, EAC separation latitude, EKE and root-mean-square error in surface fields) from the Reference state, the purpose of which was to select an appropriate perturbation. The errors grow with time, though the smaller the perturbation, the longer it takes for the model states to diverge. For example, a 1-day perturbation in initial conditions leads to the same level of error in SST after ~3-months, as opposed to after ~2-months
for a 8-days initial perturbation (results not shown).

## 2.4    The OSSEs

OSSEs were conducted to evaluate observations from several observing platforms, both existing and hypothetical, to ascertain the impact on key dynamical features. Henceforth, 'observation' refers to the synthetic observations which are derived from the free running model. We conducted 4 different OSSEs as shown in Table 1. The 'Surf OSSE' contains surface-only observations
of sea surface height (SSH) and sea surface temperature (SST), the 'XBT-N OSSE' contains the same surface observations as well as the northern XBT transect, the 'XBT-S OSSE' contains surface observations and the southern XBT transect, and the 'XBT-N+S OSSE' contains surface observations with both northern and southern XBT transects.

An example of SSH and XBT observation locations is shown in Fig. 3a. Realistic sampling patterns were used for satellite-observed SSH (e.g. all SSH observations in a 5-day DA cycle are shown in Fig. 3a), SST observations, and subsurface tem-
perature from eXpendable BathyThermograph (XBT) lines that measure temperature to 900 m in the north and south of the domain (Fig. 3a). Details on all observation types are discussed below.

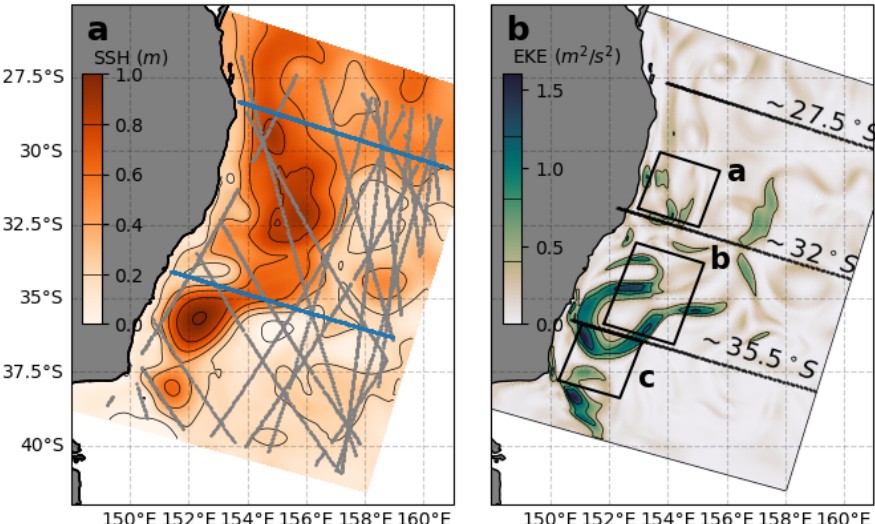

**Figure 3.** Snapshots of model conditions with examples of observations and analysis regions. (a) A snapshot of SSH (colour map) is shown with with a single 5-day cycle of SSH (grey dots) and XBT (blue dots) observations. (b) A snapshot of EKE (colour map) is shown, with boxes and lines showing key regions and diagnostic transects used in the analysis.





**Table 1.** Features of each OSSE are shown, including the model configuration and the details of the synthetic observations. Horizontal spacing of the XBT profiles are described in Sect.2.3 and spacing of typical SSH and SST data is shown in Fig. 3a. Grey fill indicates details not applicable for the forward-running Reference state.

| Experiment name | Model configuration details | Synthetic observations |
|---|---|---|
| Reference state | Free-running simulation covering period of Nov 2011–Jan 2013. Observations extracted from this simulation. | |
| Surf | 4DVar simulation covering period of Nov 2011–Jan 2013; assimilating SSH and SST 'observations' synthesised from Reference state | Along-track satellite-observed sea surface height altimetry and sea surface temperature. |
| XBT-N | Surface observations plus XBT observations along the northern transect. | XBT temperature profiles to 900 m starting at $\sim 28^\circ$ S. |
| XBT-S | Surface observations plus XBT observations along the southern transect. | XBT temperature profiles to 900 m starting at $34^\circ$ S. |
| XBT-N+S | Surface observations plus XBT observations along both transects. | XBT temperature profiles to 900 m starting at $28^\circ$ S and $34^\circ$ S. |

### 2.4.1 SSH

SSH observation timing and locations are taken from the global ocean along-track multi-mission sea level altimeter data (available at Copernicus Marine Service; https://doi.org/10.48670/moi-00146). This dataset has horizontal and along track
spacing and temporal repeats that vary between mission. For example, for the Jason-2 satellite, along-track sampling is $\sim 7$ km. Observation locations are removed in water shallower than 1000 m and near to the model domain border (within 10 cells from the eastern and southern borders and within 60 cells from the northern border), so as to reduce mismatch between boundary forcing and assimilated observations. SSH observations at the timing and along-track locations are extracted from the Reference state and the track is replicated 2 hours before and after each track, so as to inhibit the formation of barotropic waves
through the 4DVar adjustment (for more details see Levin et al., 2021). During the processing of the raw data, we generate 'super observations' by averaging multiple observations within a grid cell and within a 5 minute window, to produce a single observation value which is then provided to the Reference state simulation. A single cycle (5-day period) of SSH observations can lead to a relatively sparse SSH observation field (as per the example in Fig. 3a), especially in comparison to a daily gridded SSH observation field e.g. from AVISO (Kerry et al., 2016). Each SSH along-track observation extracted from the Reference
state is perturbed such that the perturbations have normal distribution with a standard deviation equal to the applied observation error, which for along-track SSH is 0.04 m.





### 2.4.2 SST

SST observation timing and locations are chosen to represent data sourced from the gridded, near-real time Himawari-8 satellite product. The grid for this data is time-invariant and at a higher resolution (2 km) than the model grid. As a result, we choose

each model point as an observation location. In order to reproduce realistic observation density, observations are masked by the realistic cloudiness field taken from the atmospheric reanalysis product used for surface forcing (Su et al., 2019). As there is no exact match between the BARRA-R cloudiness and the cloud matching algorithm used to process Himawari-8 data, we chose a simple cloudiness fraction (0.75) and discarded any observation locations with heavier cloud than this value. The choice of cloudiness fraction was calibrated by comparing the resulting fields against Himawari-8 images. Furthermore, we

discarded any observations at times with wind speeds less than $2 \, \mathrm{m \, s^{-1}}$ and during the day (to reduce mismatch between skin layer temperature and bulk SST, e.g. Yang et al., 2020b). Lastly, SST data near the coastline (shallower than 100 m) or within 10 cells of the boundaries is removed to avoid contamination. Observation error is set at $0.5°$C and is added as a random perturbation to the observation values. Raw SST observations are processed in the same manner described above, where observations co-located within a cell or within a 5 minute period are averaged to give a single value per cell. As the SST

data is at a similar resolution to the model grid, an example field of observations is not shown in Fig. 3a.

### 2.4.3 XBT

XBT locations were generated to very approximately match XBT deployments along two of the Scripps High Resolution XBT Program lines: PX30 from Brisbane to Noumea and the PX34 line from Sydney to Wellington. These transects are occupied nominally quarterly, with a horizontal spacing of between approximately 10 km to 100 km. In our case, synthetic temperature

transects are cross-shore, parallel to the model grid, with casts every model cell. The resultant synthetic XBT transects (Fig. 3a; blue lines) are approximately perpendicular to the EAC at $\sim 28°$ S and $\sim 35°$ S and extend to within 10 cells of the domain boundary. Each temperature profile is extracted along the line with a time gap of 30 minutes, producing a full transect in approximately 5 days. The transect is then repeated from west to east again, with a 7-day duration between the beginning of each transect. Profiles extract synthetic observations from 5 m to 900 m at the centre of each model cell (except where vertical

resolution is finer than 10 m, in which case observations are limited to be at every 10 m to match the real XBT data). While the depth extent of our synthetic observations replicates the PX30 and PX34 lines, our spatial and temporal density is much higher (approximately weekly instead of quarterly). The observation error is added as a normally distributed random perturbation with a depth-dependent profile. This error profile was developed from a climatology of depth profiles in the EAC and should capture changes in temperature variance with depth (Kerry et al., 2016). The error profile has a subsurface maximum of $0.6°$C at 300

m then decreases to $0.12°$C at 1100 m. Like with SSH and SST, the XBT observation error is applied as a random normal perturbation to the synthetic observations.





### 2.5 Key dynamic metrics

We define several key metrics, which are chosen to test the improvement in the simulation of EAC dynamics, as ultimately this is the area of interest for improved prediction. We will assess quantities that are relevant for the key uncertainties highlighted
in Sect. 1, namely, ocean heat content at different depths, spatial localisation of the EAC jet and the characteristics of the EAC eddy field.

#### 2.5.1 Mean Kinetic Energy (MKE) and Eddy Kinetic Energy (EKE)

MKE and EKE have been used previously for model validation of the EAC jet and the EAC eddy field (e.g. Li et al., 2021). Here, we use EKE and MKE as metrics for assessing how well our OSSEs represent the EAC and EAC eddy field. For
flow components in the zonal and meridional directions ($u$ and $v$, respectively), we calculate MKE as half of the sum of the squared time-mean velocity components, as in MKE $= \frac{1}{2}(\bar{u}^2 + \bar{v}^2)$. The EKE is then half of the sum of the squared-anomaly of the velocity components from the long-term mean, as in EKE $= \frac{1}{2}(u'^2 + v'^2)$. The time-mean zonal ($\bar{u}$) and meriodional ($\bar{v}$) components are averages of the instantaneous velocities over the full model integration (November 2011 to January 2013), as in $u' = u - \bar{u}$, with $v'$ defined likewise. Note that all kinetic energies given as values per unit mass. We explore these metrics
at the surface and at $500$ m , below the typical depth of the core of the EAC jet.

#### 2.5.2 Root Mean Square Error

The root-mean-square error (RMSE) is calculated as $RMSE = \sqrt{\overline{(\hat{X} - X)^2}}$, where the time mean (shown with an overbar) is taken of the difference between a reference field $\hat{X}$ and the quantity in question $X$, at each grid point. Here we apply this standard definition to compute a temporally-averaged RMSE, by taking the mean of the difference at every model output time
separately for each model cell.

#### 2.5.3 Upper Ocean Heat Content (UOHC)

We calculate a volume-integrated Upper Ocean Heat Content (UOHC) by taking the depth and spatial integral of temperature, scaled by the specific heat capacity ($c_p = 3850$ J kg$^{-1}$ °C$^{-1}$ ) and reference density ($\rho_0 = 1026$ kg m$^{-3}$ ), as in

$$\text{UOHC}(t) = \int\limits_{\eta_1}^{\eta_2} \int\limits_{\xi_1}^{\xi_2} \int\limits_{0}^{H} \rho c_p T(x, y, z, t)\, dz\, dx\, dy,$$

where the integrals extend from the surface to a specified depth ($H$), and over the region $\xi_1$ to $\xi_2$ in the model-space $x$-direction and $\eta_1$ to $\eta_2$ in the model-space $y$-direction. The regions chosen are specified with model-space coordinates (rather than geographic coordinates) so that they are roughly aligned in the cross- and along-shelf directions.

The integrated UOHC is calculated for three regions. The first region (region a in Fig. 3b; centered at 154° E, 31.3° S) covers the EAC jet, upstream of the typical EAC separation zone. This region is chosen to capture heat in the EAC core, typically
before it meanders and forms eddies. The second box (region b in Fig. 3b; centered at 153.5° E, 34.6° S) is aligned over the





EAC eddy field and separation zone, in a region identified to have the highest surface magnitudes and variability in EKE as per (Li et al., 2021). The third box (region c in Fig. 3b; centered at 151.5° E, 37° S) is located over the EAC southern extension (see Oke et al., 2019), in a region identified to have the highest rate of UOHC warming over the two decades from 1996–2016 (Li et al., 2022a), and so is an important region to represent correctly in models.

## 3 Results

### 3.1 Spatial representation of the EAC

Key circulation features of the EAC, which emerge in the long-term average (Oke et al., 2019), include the EAC southern extension (which continues south following the shelf), the EAC return flow (which heads north on the eastward side of the EAC retroflection), and the EAC eastern extension (which flows eastwards towards New Zealand). All of these features are observed in the time-mean MKE of the Reference state (Fig. 4a at the surface and to a lesser extent at 500 m depth in Fig. 4f).

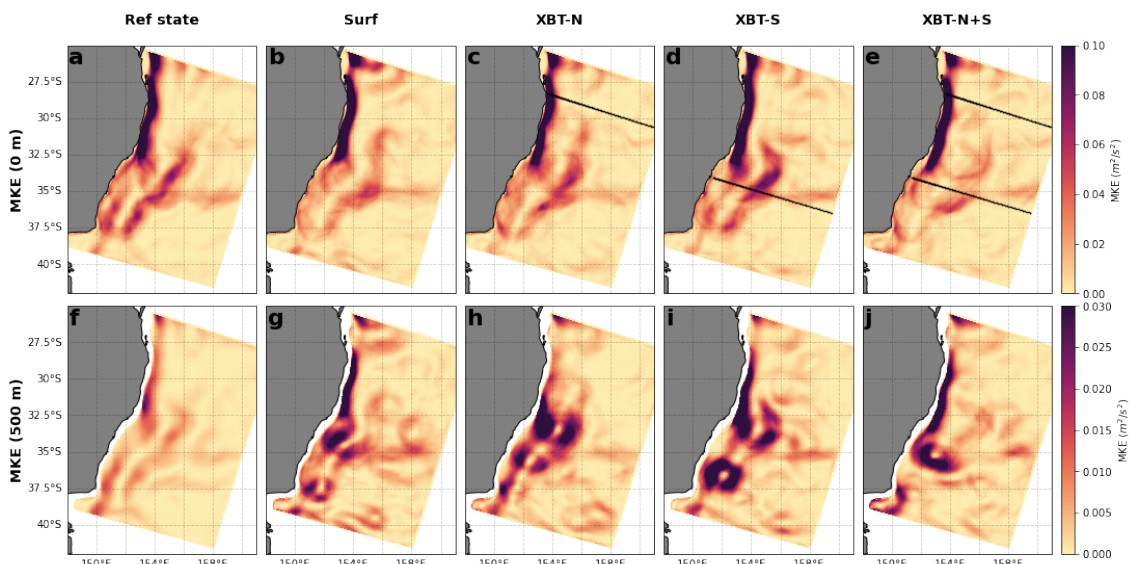

**Figure 4.** The surface MKE for the (a) Reference state, and OSSEs (b) Surf, (c) XBT-N, (d) XBT-S, (E) XBT-N+S, and the MKE at 500 m for the (f) Reference state, and OSSEs (b) Surf, (c) XBT-N, (d) XBT-S, (E) XBT-N+S. Black lines in panels b-d indicate location of the XBT lines.

The surface-only OSSE (Surf) has reduced energy through the EAC return flow region, that is, south of 34° S (Fig. 4b). Other features are fairly well represented. Both of the northern and southern XBT OSSEs represent the surface MKE of the Reference state well, with a clear jet, separation region, return flow and two bands of eastern extension (Fig. 4c-d). The XBT-N+S experiment has poor representation of the return flow, but, unlike the surface-only experiment, poorly captures the eastern extension (Fig. 4e).





In the Reference state, at 500 m, the magnitude of MKE is substantially reduced but dominated by the jet region and the return flow (Fig. 4a,f). At this depth, there are more significant changes to the MKE distribution in the OSSEs, which all overestimate the magnitude of MKE at 500 m, a demonstration that reanalyses struggle to resolve conditions at depth. The surface-only observations do not constrain MKE well at 500 m depth, leading to anomalously deep re-circulation features

throughout the domain (Fig. 4g). The addition of temperature observations along the northern transect (Fig. 4h) produces a better representation of the return flow, but re-circulation features form along the EAC (for example at 32.5° S), without feeding any eastward flow. The XBT-S experiment (XBT-S; Fig. 4i) best reproduces the Reference state MKE pattern, with a clear return flow and eastern extension, though the magnitude of MKE at 500 m is too high (as in all the OSSEs). When both XBT transects are present (Fig. 4j), MKE is again poorly represented, with re-circulation too far north in the Tasman Sea (at

35° S).

The mean SSH is represented similarly in all experiments, likely because SSH observations are assimilated in all OSSEs (see Fig. B1a). It also indicates that subsurface observations have little impact on the representation of SSH (see Fig. B1b-e). The representation of the EAC eddy field, captured in EKE at the surface and 500 m is also similarly represented in all OSSEs (see Fig. B1f-o).

**3.2   Spatial representation of temperature**

The surface temperature conditions are dominated by the extension of warmer water carried southwards by the EAC (Fig. 1). As all OSSEs assimilate the same SST data, the differences in representation of surface temperature are negligible (not shown). Instead, the spatial differences in representation of temperature fields below the surface are shown at 250 m, 500 m, and 1000 m in Fig. 5.

The error in temperature at 250 m is much higher for the Surf OSSE (Fig. 5b), indicating that surface observations alone are not enough to constrain conditions at intermediate depths, and actually degrade temperature representation at 250 m and 500 m, but by 1000 m depth the error is reduced. The presence of the subsurface observations greatly improves 250 m temperature RMS (Fig. 5c-e). Assimilating the southern XBT transect removes the band of high RMSE in the EAC eddy field at 34° S which is dominant in the Surf OSSE and the northern XBT transect OSSE (cf. Fig. 5d to Fig. 5b-c). Note the improvement

in RMS error upstream of the XBT-S transect location, compared to surface-only observations. The addition of the north and south transects in the XBT-N+S OSSE (Fig. 5e) improves temperature RMS compared to the Surf OSSE and the XBT-N OSSE, and has similar spatial patterns in RMS error to the southern XBT OSSE.

At 500 m, the error in temperature is generally slightly lower for each OSSE compared to the 250 m temperature field, though the reduced variability at depth likely partly accounts for this (compare contours of standard deviation in the temperature

of Reference state at 250, 500 and 1000 m; Fig. 5a,f,k). The OSSEs with either a northern or southern transect of XBT observations (Fig. 5h-i) display relatively low RMS error, especially compared to the surface only observations (Fig. 5g). The addition of both transects together (Fig. 5j) slightly increases RMS error in the separation zone.



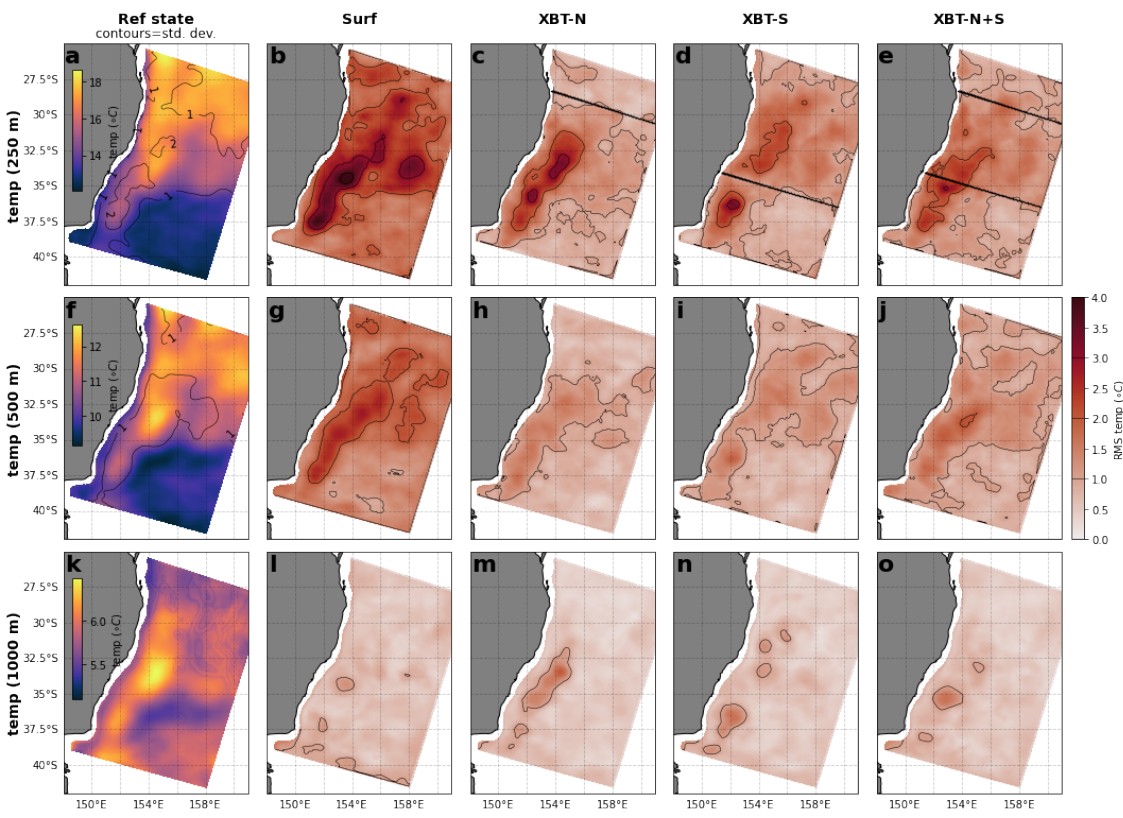

**Figure 5.** The differences in spatial representation of temperature between the Reference state and each OSSE. (a) The mean temperature field at 250 m in the Reference state, with contours of standard deviation in that field. The RMS error between the Reference state and the OSSEs (b) Surf, (c) XBT-N, (d) XBT-S, (e) XBT-N+S. The second row (panels f-j) is the same, but at 500 m, and the third row (panels k-o) shows the same fields at 1000 m. Note that colour scales are identical for all RMS error plots. In Reference state panels, contours are the standard deviation in the respective field while in all OSSE panels, the contours are of RMSE. Contour intervals are the same for all panels and are shown as a line in the RMSE colourbar. Black lines in panels c-e indicate location of the XBT lines.





Temperature at 1000 m is represented with similar RMS error in all experiments (Fig. 5l-o). The lower temporal variability (Fig. 5k) in temperature at this depth likely leads to good representation from the boundary forcing conditions for all experi-

ments.

## 3.3 Representation of velocity

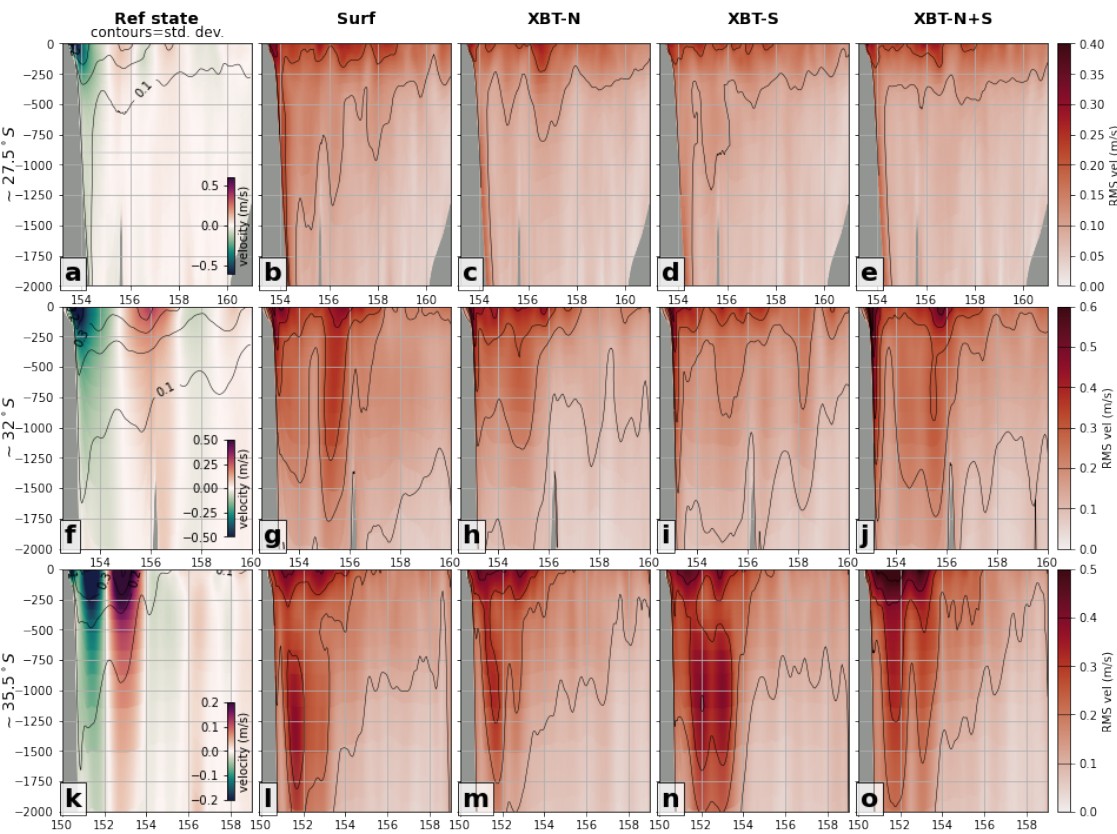

**Figure 6.** Vertical cross-sections of meridional (north/south) velocity at key transects (first row) ∼27.5° S, (second row) ∼32° S and (third row) ∼35.5° S are shown for each experiment. (a) The time-mean velocity (northwards-positive) in the Reference state, with contours of standard deviation in the Reference state velocity. The RMS error between the Reference state and the OSSEs (b) Surf, (c) XBT-N, (d) XBT-S, (e) XBT-N+ is shown at the transect ∼27.5° S. The second row (f-j) shows the same fields as the first row, but for the second transect at ∼32° S, while the third row (k-o) shows the same fields but for the third transect at ∼35.5° S. In Reference state panels, contours are the standard deviation in the respective field while in all OSSE panels, the contours are of RMSE. Contour intervals are shown as a line in the RMSE colourbar.

Simulated northwards velocity (positive northwards) at key transects are best resolved with the addition of a subsurface observations (Fig. 6). In the upstream region, the EAC core (strong southward flow above 250 m between the coast and 154° E; Fig. 6a) is a region of higher RMS error in all OSSEs (Fig. 6b-e), but is improved with subsurface observations (e.g. compare





Fig. 6b to Fig. 6c-e). Notably, the presence of the southern XBT line reduces RMS velocity error in the EAC jet and on the slope (Fig. 6d) to levels comparable to the RMS for the northern XBT line (Fig. 6c).

In the separation region, the mean southward EAC (from the coast to $155°$ E) and the northward return flow between $155°$ E–$158°$ E (Fig. 6f) are regions of high RMS error in the surface-only (Fig. 6g) OSSE. The southern XBT line OSSE (Fig. 6i) reduces this error within the southward EAC flow and return flow, particularly between 200–1500 m; notably RMS error below

the deepest XBT observation (that is, deeper than $\sim 900$ m) is also reduced compared to the surface-only experiment. When both transects are present (Fig. 6j), RMS error on the shelf break and deeper than 1000 m increases compared to either OSSE with only a single observation transect.

In the downstream eddy-rich region, RMS error is generally high in all OSSEs, as would be expected in this dynamic and less predictable region. The addition of temperature observations at depth does little to improve meridional velocity RMS as

compared to surface only observations.

### 3.4 Representation of vertical temperature structure

The time-mean temperature at three cross-slope transects located upstream, near the separation zone and downstream in the high EKE region (see Fig. 3b for locations) is shown in Fig. 7. These transects are not co-located with the XBT lines. The Reference state shows cooler temperatures further south as well as the isotherm deepening associated with warm core eddy

downwelling (first column).

For the upstream transect ($\sim 27.5°$ S; Fig. 7a), RMS error in temperature is improved when depth observations are included, as compared to the Surf OSSE (cf. Fig. 7c-e to Fig. 7b). The improvement in the representation of the EAC jet core temperature (i.e. temperature in the region between the coastline and $155°$ E, above $\sim 1000$ m) is present for all OSSEs. While the lowest RMS error along this transect is produced by the northern XBT transect OSSE (which is understandable given the proximity

of the sampled transect and the observations), the southern transect OSSE also has far lower RMS error than the Surf OSSE, despite the southern XBT observations being $\sim$600 km distant.

Nearer to the separation zone ($\sim 32°$ S; Fig. 7f), RMS error for the Surf OSSE is higher, especially in the mid depths of 250 m –500 m (Fig. 7g), with this region of higher error stretching down to 1000 m. The addition of subsurface observations in the north does improve this error somewhat (Fig. 7h). The presence of the southern XBT transect further reduces RMS error near

the separation zone (Fig. 7i), indicating an improvement in representation of heat content carried by eddies.

In the eddy-rich region ($\sim 35.5°$ S; Fig. 7k), the region of greatest RMS error is again the 250 m –500 m depths at $152°$ E. Both single transect experiments retain a relatively high RMS error, often extending to 1500 m. The XBT-N+S OSSE has relatively good representation of temperature along this transect and much improved representation below 1000 m (Fig. 7o).

### 3.5 Representation of upper ocean heat content

Volume-integrated upper ocean heat content, in three regions of interest and across the top 700 m and 2000 m, is best simulated with the addition of subsurface observations (Fig. 8). The temporal evolution of UOHC for the Reference state (Fig. 8; solid



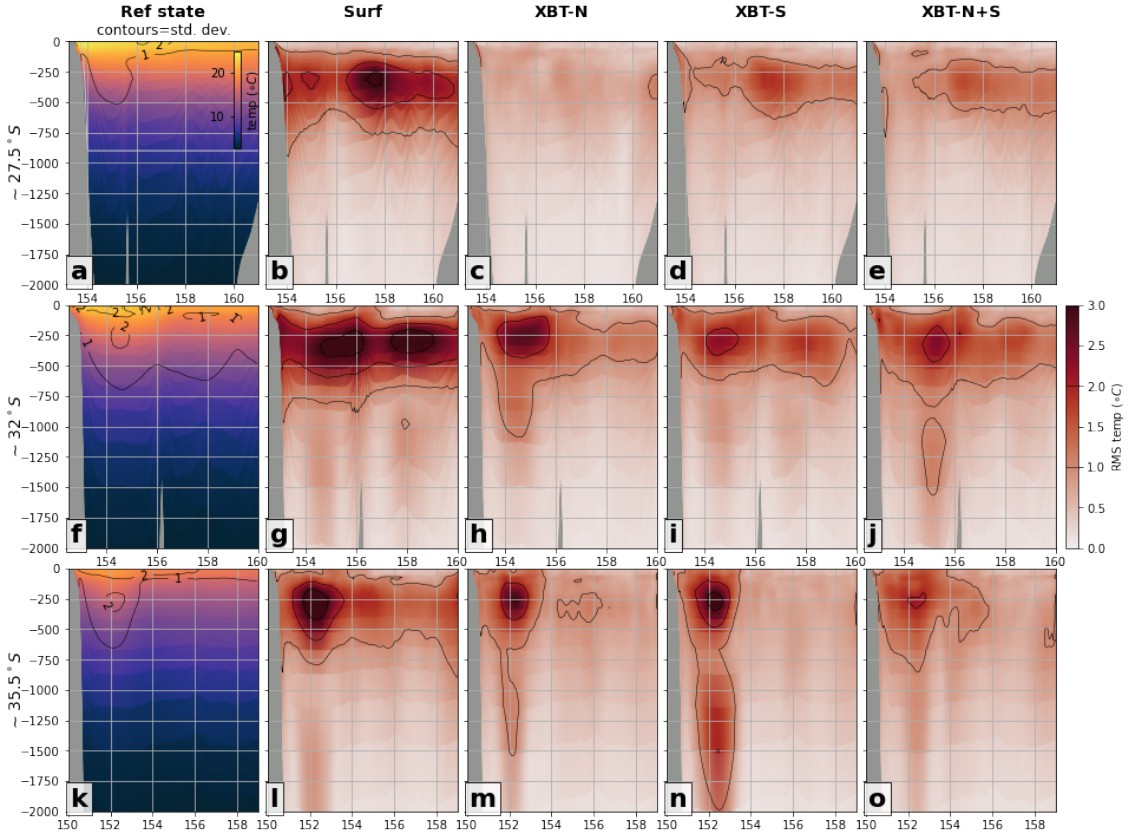

**Figure 7.** Differences in vertical transects of temperature for each experiment are shown for the ∼27.5° S, ∼32° S and ∼35.5° S transects. (a) The mean temperature of the Reference state with contours of standard deviation. The RMS error between the Reference state and the OSSEs (b) Surf, (c) XBT-N, (d) XBT-S, (e) XBT-N+S is shown at the ∼27.5° S transect. The second row (f-j) shows the same fields as the first row, but for the second transect at ∼32° S, while the third row (k-o) shows the same fields but for the third transect at ∼35.5° S. In Reference state panels, contours are the standard deviation in the respective field while in all OSSE panels, the contours are of RMSE. Contour intervals are shown as a line in the RMSE colourbar.

blue line shallower than 700 m, and dashed blue line, shallower than 2000 m) is shown in comparison to each OSSE. The RMS errors in UOHC between each OSSE and the Reference state is shown in Table C1.

In the region immediately upstream of the typical EAC separation zone (Fig. 8a; see box a in inset), all OSSEs represent UOHC content relatively poorly (Fig. 8a; Reference state is blue line). For the upper 700 m, the XBT-N transect best represents UOHC ($\mathrm{RMSE} = 0.090$ ZJ ; Table C1) while for the upper 2000 m the best representation of UOHC is in OSSEs that contain either the XBT-N or XBT-S transects (which both share $\mathrm{RMSE} = 0.13$ ZJ ; Table C1). It is noteworthy that the Surf OSSE consistently gives the worst representation of UOHC in all three boxes (with RMSE ranging from 0.18–0.35 ZJ for the upper 700 m and RMSE ranging from 0.18–0.27 ZJ for the upper 2000 m ; see Table C1).





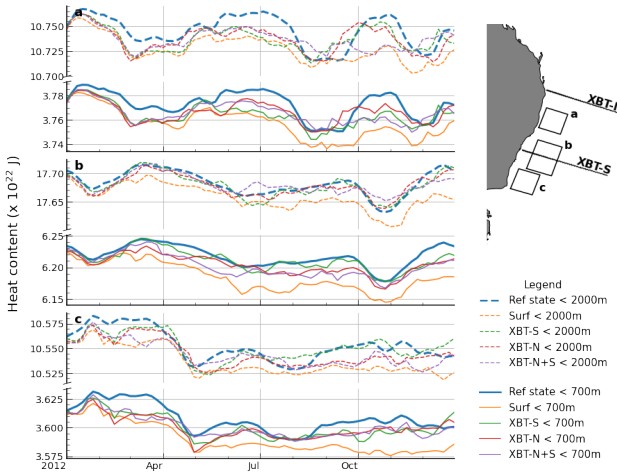

**Figure 8.** The temporal evolution of integrated upper ocean heat content for three key regions: (a) upstream of the EAC separation zone, (b) the high EKE variability region, and (c) the region with strong UOHC trends, with locations shown in inset. The upper 700 m is shown with solid lines and the upper 2000 m is shown with dashed lines. Note the broken y-axes.

This improved representation of UOHC in OSSEs with either XBT-N or XBT-S is again featured in the high EKE variability region (Fig. 8b; see box b in inset). Including the southern transect (XBT-S; Fig. 8b green line) gives the best representation shallower than 700 m (RMSE = 0.093 ZJ ; Table C1), though this is somewhat expected given that XBT-S passes through this region. UOHC for the upper 2000 m is equally well represented by either XBT-N or XBT-S (RMSE = 0.083 ZJ ; Table C1).

    In the region of greatest upper UOHC trends (Fig. 8c; see box c in inset) as identified in Li et al. (2022a), we again see the
best match between the Reference state and the XBT-S OSSE, especially for the upper 2000 m (RMSE = 0.086 ZJ ; Table C1). It is notable that the southern XBT transect improves UOHC upstream (in box a) and downstream (in box c) of its location, as well as down to depths of 2000 m , well below the deepest XBT observation.

### 3.6   Observation impact during different EAC phases

    When the EAC sheds large anticyclonic eddies, there is a rapid retraction of the EAC separation latitude. Typically, this
EAC separation region is between $31°$ S to $32°$ S (e.g. Cetina-Heredia et al., 2014), however, separation can occur north or south of this typical separation zone. As the presence of the EAC jet or large anticyclonic eddies will change meridional heat supply, downstream conditions will be different during northern or southern separation phases. Consequently, fixed location observations may have a different impact on representing the EAC and eddy field during different EAC phases. We have chosen two 2-month periods that represent 'northern separation' (01 September 2012 to 01 November 2012) and 'southern separation'
(01 March 2012 to 01 May 2012) phases. Selected metrics are then compared between experiments within and between these northern and southern separation phases to illustrate the differences in observation impact.





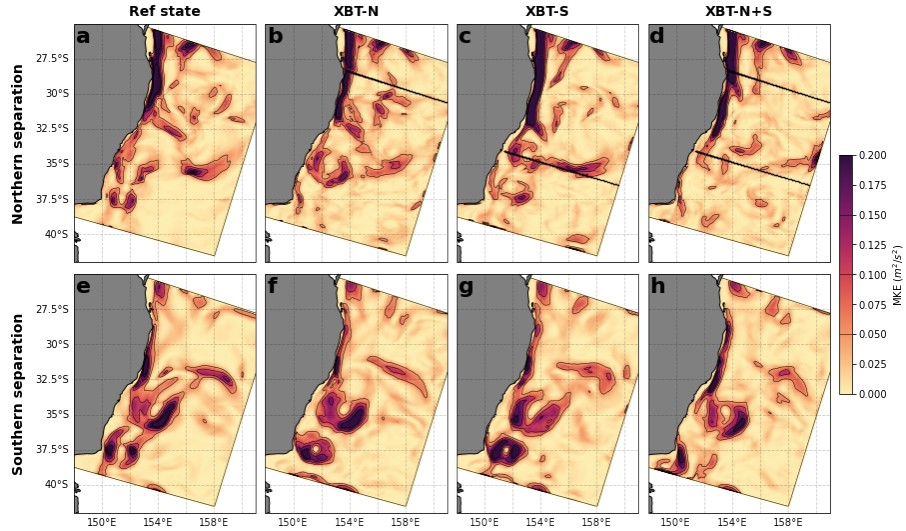

**Figure 9.** The surface MKE is compared between the (a) Reference state, (b) XBT-N, (c) XBT-S and (d) XBT-N+S for the case when the EAC is in a northern separation phase. For the southern separating EAC phase, the surface MKE is compared between the (e) Reference state and the (f) XBT-N, (g) XBT-S, (h) XBT-N+S. Black lines in panels b-d indicate location of the XBT lines.

For the period 01 September 2012 to 01 November 2012, the EAC separated further north than the mean, at $\sim31.5°$ S (Fig. 9a) in the Reference state. As we have established in Sect. 3 that including subsurface observations provides the best representation of EAC circulation and vertical structure, we exclude the Surf OSSE and show only the OSSEs that include
subsurface observations together with SSH and SST observations.

For this period MKE is best represented in the experiments with either the northern or southern subsurface observation transect (cf. Fig. 9a and Fig. 9b-c). Only the XBT-N and XBT-S experiments have the separation and flow into the eastern extension at the correct location (with MKE connecting from the northern separation point to 156° E, 35° S) and with realistic magnitude. Adding both XBT transects produces separation further south and reduced MKE in the eastern extension (Fig. 9d).

Over the period 01 March 2012 to 01 May 2012, the EAC separated further south, with a separation at 35° S and re-attachment feeding the southern extension and initiation of the return flow at 38° S (Fig. 9e). When the EAC separates further south, there is a connected band of MKE along the coastline, re-circulation features at 35° S and 38° S, a strong northward return flow and an eastern flow extension. The magnitude of MKE in the jet is weaker than when the EAC separates further north. Both XBT experiments better represent these flow features along the coast and the northward return flow (Fig. 9f-g).
The XBT-N+S OSSE does not represent the full extension of the EAC nor the return flow as well (Fig. 9h).

The northwards velocity during each separation phase and the error in representation displayed by each OSSE is shown in Fig. 10. During northern separation, the jet region (Fig. 10a; upper 500 m west of 155° E) and surface ocean to a depth of 250 m are the sources of greatest error in the upstream region, for all OSSEs (Fig. 10b-d), however observations in this upstream region (Fig. 10b,d; XBT-N and XBT-N+S) lead to the largest improvement in error in representation of the EAC jet. During



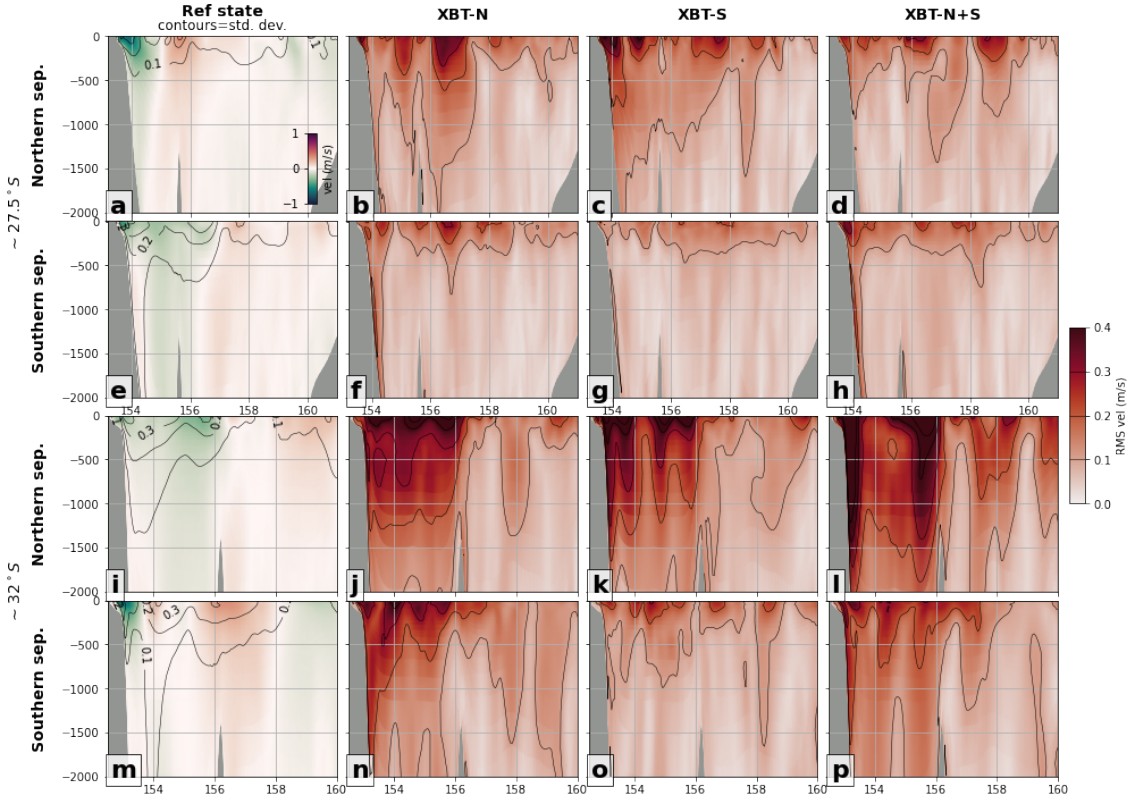

**Figure 10.** The northwards velocity at $\sim$27.5° S is compared between the (a) Reference state, (b) XBT-N, (c) XBT-S and (d) XBT-N+S for the case when the EAC is in a northern separation phase, and for the (e) Reference state, (f) XBT-N, (g) XBT-S) and (h) XBT-N+S in the southern separation phase. Likewise, the northward velocity is compared along the transect at $\sim$32° S for the (i) Reference state, (j) XBT-N, (k) XBT-S and (l) XBT-N+S OSSEs in the case when the EAC is in a northern separation phase. Lastly, velocities along the same transect in the (m) Reference state, (n) XBT-N, (o) XBT-S and (p) XBT-N+S OSSEs are compared for the southern separation phase. In the Reference state panels, contours are the standard deviation in the respective field while in all OSSE panels, the contours are of RMSE. Contour intervals are shown as a line in the RMSE colourbar.

southward separation, the southern transect of subsurface observations produces the best representation of upstream velocity, including of the EAC jet (more so than northern XBT observations; cf. Fig. 10f and Fig. 10g).

  At $\sim$32° S (Fig. 10i), both experiments with northern subsurface observations present (XBT-N and XBT-N+S, Fig. 10j,l) have higher error than with southern subsurface observations (Fig. 10k), but in general, velocity representation during a northern separation has the poorest representation, likely due to the dynamic features present in this region.

During southern separation, velocity representation along the $\sim$32° S transect is better in the presence of southern rather than northern subsurface observations (compare Fig. 10o to Fig. 10n). This is another demonstration of the high observational impact of measurements taken in the eddy-rich region. The XBT-N+S OSSE has relatively high error in northwards velocity in





this region (Fig. 10p), possibly because the DA scheme is forced to minimise the error at both the north and south subsurface
observations, resulting in a slightly degraded fit to either individually. Notably, at either transect, there is lower RMS error in
representation of velocity in all experiments, when the EAC separates further north (e.g. compare Fig. 10b-d and (Fig. 10f-h).

## 4    Discussion

### 4.1    Subsurface observations improve key EAC features

While the MKE of the EAC jet is well reproduced with surface-only observations, observations at depth are required to correctly
represent other important EAC features. In particular, assimilating subsurface temperature (XBT) observations improves the
representation of the northward return flow. The EAC eastern extension, which directs flow towards New Zealand (via an eddy
train, see Oke et al., 2019), is best represented with the inclusions of temperature measurements through the southern section.
Furthermore, only the experiment with subsurface temperature observations in the southern high EKE region reproduces these
features at depth.

The better reproduction of key EAC features in the MKE fields (Fig. 4) by assimilating temperature observations at depth
(together with surface observations) is due to an improvement in estimates of subsurface conditions. This highlights that correct
density structure is important for resolving features such as the return flow and the EAC eastern extension, and geostrophic
balance (i.e. conditions inferred from surface observations only) is not fully representative of flow in these features.

Other studies have also emphasised the importance of high resolution and subsurface observations. In the Kuroshio Current,
OSSEs showed that density changes resulting from subsurface temperature and salinity observations had a larger impact on
WBC circulation than on deep, open ocean circulation (Lee et al., 2020). OSSEs conducted in the Gulf of Mexico found that
higher spatial resolution subsurface temperature and salinity observations substantially reduced error by representing variability
at scales too small to resolve by surface-only observations (Halliwell et al., 2015).

Surface EKE is not further improved with the assimilation of temperature observations at depth. Furthermore, the improve-
ment from temperature (XBT) observations on EKE at depth ($500 \, \text{m}$) is also minimal (there is some improvement from the
XBT-S transect, though this includes degradation in the far south of the domain). The limited impact is likely because EKE
is a measure of the fast time-scale fluctuations in velocities, and temperature at depth has limited correlation with these rapid
velocity changes. This is in contrast with MKE, as the mean flow is highly dependent on the subsurface temperature structure.

Observations of subsurface temperature at either XBT transect location vastly improve simulated temperature at $250 \, \text{m}$ – $500$
$\text{m}$ compared to surface-only observations (although the southern XBT has more impact; see Sect. 3.4). This is consistent with
Siripatana et al. (2020) who showed that the assimilation of subsurface observations (from a deep-water mooring array and
ocean gliders) where required to correctly represent the depth of the EAC core and eddies. Improvement in the temperature
field resulting from the subsurface observations also extends far upstream of the observation transect, indeed, 'downstream'
observations at 34° S improve representation northward, all the way to 27.5° S, over $600 \, \text{km}$ distant (Fig. 5d). This far-reaching
impact was demonstrated in this 4DVar configuration of the EAC system by directly computing observation impact (Kerry et al.,





2018), for example temperature measurements from ocean gliders off 34° S were shown to impact estimates of EAC transport upstream at 27.5° N.

The vertical structure of eddies is complex, making them difficult to model. In this study, the greatest errors in representation are in the shallow-intermediate waters from 250 m –500 m (Fig. 7), particularly in the eddy-rich southern regions. Assimilating data along the northern XBT transect improves the vertical temperature structure in the upstream, shallow-intermediate waters,

including in the EAC core (Fig. 7c), but does little to improve error in the temperature structure of more eddy-rich regions (cf. Fig. 7c and Fig. 7h,m). In contrast, including data from the southern XBT line improves representation of temperature both upstream and in the more eddy-rich regions of the typical EAC separation zone, and to a lesser extent, the southern Tasman Sea (Fig. 7d,i,n). While it is not surprising that temperature observations at depth improve the vertical temperature structure, it is notable that improvement extends to a depth of 1250 m , which is well below the deepest observation (900 m ). Improvement

below the depth of the observation transect also occurs for the meridional velocity. This suggests that conditions (temperature and velocities) at intermediate depths co-evolve with conditions above.

## 4.2   Subsurface temperature observations improve UOHC representation

Given the improvement in representation of temperature and velocities at depth, due to the inclusion of two transects of subsurface temperature observations at relatively fine spacing, we would expect a marked improvement in the representation

of UOHC. Indeed, we see that at all chosen locations, the inclusion of only surface observations in the Surf OSSE leads to the poorest representation of UOHC compared to the Reference state (Fig. 8), in contrast to when subsurface observations are included. This is true for both 0–700 m and 0–2000 m integrated UOHC.

In particular, assimilating the XBT-S observations leads to the closest match with the Reference state in the upper 700 m in the high variability region (Fig. 8b) and in the region identified in the southern Tasman Sea as having large UOHC trends

(Fig. 8c). This southern XBT transect also improves UOHC both upstream (Fig. 8a) and downstream (Fig. 8c), indicating the far reaching impact of observations in this particular region.

Behrens et al. (2019) demonstrated a strong relationship between UOHC anomalies and the location, occurrence and intensity of MHWs. In the southern Tasman Sea, a strong positive trend in EKE is also shown to be linked to a positive trend in UOHC (Li et al., 2022a). As a result, high-resolution subsurface temperature observations should improve representation of

UOHC in model analyses, leading to improved predictability of MHWs. As MHWs can have significant ecological impacts, this strongly motivates the inclusion of subsurface observations in operational forecasts.

## 4.3   Downstream subsurface temperature observations are more impactful than upstream observations

For the many metrics that we have explored here, observations from the XBT-S transect were highly effective at assimilating towards the Reference state, for example, MKE and EKE at 500 m as well as ocean heat content. This finding agrees with

other literature which has suggested at the importance of observing in high variability regions. Using an observational impact analysis, Kerry et al. (2018) found that observations in the region of high natural variability from 32° S–37° S have the greatest impact per observation on transport estimates. In the Solomon Sea, Melet et al. (2012) suggested that gliders (which mea-





sure temperature and salinity) piloted through regions of high variability had a strong impact on reducing error in modelled thermocline properties.

The southern transect of subsurface observations improves the representation of conditions (e.g. horizontal and vertical temperature distribution) ∼600 km upstream (as well as in the region downstream of the separation zone and in the EAC eddy field). This is in contrast to the northern transect, which does not significantly improve conditions downstream of the separation zone. Possibly, this is because observing in a region where instabilities are beginning to grow constrains the conditions that led to eddy formation upstream, and is a tighter constraint on eddy evolution. For example, Li et al. (2021) identify a band of strong

barotropic energy conversion (mean flow shear-induced eddy generation) between $31.5°$ S and $33.5°$ S, which is immediately north of the XBT-S observation transect.

    In contrast, sampling upstream improves the EAC jet representation, but provides limited constraints on conditions after the EAC loses coherency and sheds eddies. This further illustrates that the depth structure of eddies is not well constrained by conditions in the upstream jet. Comparing the results of DA experiments including different observations, Siripatana et al.

(2020) also showed that the constraint from an upstream, deep-water mooring array on the EAC core depth is effective where the jet is typically coherent, but degrades downstream in the eddy-dominated region.

### 4.4   Impact of EAC phase (separation latitude) on error in state estimates

The impact of surface-only and subsurface observations on the representation of EAC circulation features (e.g. eastern extension and return flow) and the vertical velocity structure differs depending on whether the EAC is separating to the north or south

of the typical separation latitude range. The mean separation latitude ranges between $31–32°$ S (Cetina-Heredia et al., 2014; Oke et al., 2019); we identify northern separation as occurring between $28–31°$ S, and southern separation occurring south of $32°$ S (results not shown). A separation to the north has been shown to be associated with a high energy EAC jet which becomes unstable and separates from the coast to the north of the typical separation latitude (Li et al., 2021). This northern separation results in reduced heat transport to the south of the domain (as shown in Fig. 8, for example in October-September, by the

reduced UOHC in box b and increased UOHC in box a). Conversely, a southern separation indicates a lower energy EAC jet, more stable mean flow upstream, and increased heat transport downstream. This is consistent with previous studies (Li et al., 2022a, 2021) who showed a low frequency relationship between upstream EAC energy, separation latitude and downstream UOHC. The results of Li et al. (2022a) were derived over interannual periods; we confirm this relationship for two specific 2-month periods in 2012.

When the EAC is separating to the north of the typical separation latitude range, representation of circulation and vertical conditions, (that is, the state estimate) have relatively high error. This is in contrast to a more southern separation, where the representation of these features is achieved with less error. For example, errors in estimates of the vertical structure of the EAC and eddy field are lower (compare RMS error between Fig. 10b-d and Fig. 10f-h). Lower errors during southern separation is because the more stable EAC is less prone to instabilities which inhibit model skill.

During a northern separation phase, experiments with either line of subsurface observations (that is, XBT-N and XBT-S) produce similar representations of the EAC circulation and flow at depth (cf. Fig. 10b and Fig. 10c). Conversely, during





southern separation, observations along the southern XBT line produce less error and a better estimate of the MKE and vertical structure of velocity (cf. Fig. 10f and Fig. 10g). The more accurate state estimates produced with downstream observations during a southern separation phase is another example of the impact of sampling in high variability regions. However, the

reduced differentiation in model skill between XBT-N and XBT-S during a northern separation phase could result from several factors. The relative impact of either upstream or downstream observations may be balanced, because upstream observations are geographically closer to the separation region, while downstream observations are further south of the typical separation region. Together with the reduced coherency of the EAC making state estimates in general more difficult, this may act to equalise the impact of downstream or upstream observations on estimates of EAC conditions during a northern separation

phase.

Observations and models suggest an intensification and southward migration of the extent of the EAC (e.g. Cetina-Heredia et al., 2014; Hu et al., 2015) likely driven by a poleward shift of the South Pacific subtropical gyre Yang et al. (2020a). Our two case studies show that a southward shift in the southern extension of the EAC, co-occuring with a weaker and more stable EAC system, should be able to be represented in DA models with lower error. Furthermore, if these long-term changes in the

latitudinal extent of the EAC and EAC extension are as a result of more periods of lower MKE through the EAC jet, this could lead to more accurate simulation of downstream OHC and OHC extremes (such as Marine heatwaves).

### 4.5 Implications for future observing system design

The experiments presented in this study are well suited for providing recommendations for future observing system design. As we show, subsurface temperature measurements improve the representation of: the EAC structure including the return flow and

eastern extensions; the spatial and vertical distribution of temperature within the ocean; vertical velocity shear and therefore, ocean heat content. Kerry et al. (2018) also demonstrated the importance of subsurface observations, in particular, velocity and temperature observations at depth from deep mooring arrays and glider transects. The EAC transport array constrains velocities in the most coherent upstream EAC region, while gliders provide high spatial and temporal observation density in the eddy rich region that improve EAC transport and EKE. While we found that subsurface XBT observations were not significantly better

at simulating EKE than just surface observations, we do see a difference below the surface. Our results suggest that subsurface temperature observations from XBT transects, given appropriate temporal and spatial density, and deployed in regions of high variability, can also improve representation of the EAC jet and other EAC features both up and downstream.

Observations in the region of high variability and high EKE, south of the typical EAC separation zone (e.g. XBT temperature transect at 35.5° S), are shown to disproportionately improve representation of the EAC, compared to just surface observations

or to subsurface observations upstream. For this reason, observations taken in the high EKE region of the Tasman Sea will have a high cost-effectiveness in improving simulations of the EAC broadly. Along similar lines, O'Kane et al. (2011) speculate that observations targeting regions of large coherent forecast errors (e.g the Tasman Sea) would improve subsequent forecasts.

Another interesting implication from this study is the varying impact of subsurface temperature observations during different dynamical regimes of the EAC; i.e. northern and southern separation phase. Hence, a sampling strategy that adapts to the

changes in separation latitude can exploit the lower state estimate error during a low energy EAC phase. For example, during a





high MKE, northern EAC separation phase, high density sampling could be used to capture MKE, vertical structure and eddy shedding; this could be offset by reduced sampling in a low MKE, southern EAC separation phase.

These results suggest at future research that would further explore the subsurface representation of dynamic features and the cost-effectiveness of observing platforms in the EAC. For example, how does the simulated EAC jet and downstream

features degrade as the (spatial and temporal) resolution of subsurface observations are degraded. Existing observing platforms of this style (high spatial resolution subsurface observations such as XBT transects) could also be assessed, and the spatial and temporal footprint of their impact could be measured.

## 5  Conclusions

The East Australian Current (EAC) is the western boundary current of the South Pacific subtropical gyre and dominates the

oceanographic conditions in the Tasman Sea. Predictability of this current system is inhibited by the dynamic nature of the EAC eddy field. Using observing system simulation experiments, we explored how sea surface temperature and sea surface height observations in combination with subsurface temperature observations improve the representation of key EAC features in model state estimates. Improving model state estimates is key to improved prediction.

While surface observations are effective at improving representation of key EAC circulation features at the surface, such as

the return flow and the southern and eastern extensions, surface observations struggle to represent these features at depth - hence subsurface observations are critical for improving representation at depth. In particular, the northward return flow and eastern extension are represented only when including subsurface observations in the south of the domain. Not only is the vertical temperature structure improved with subsurface temperature observations, but so is the vertical velocity structure. As a result, upper ocean heat content, at key locations along the eastern seaboard, is best represented by assimilating subsurface temperature

observations. Given the link between upper ocean heat content anomalies and marine heatwaves, improving representation of ocean heat content in simulations is a priority.

We demonstrate that observations taken in the high EKE region of the Tasman Sea have a strong impact on improving representation of the EAC and its vertical structure - upstream and downstream of the observing location. We posit that observing in the location where instabilities are growing into eddies gives information to the assimilation system of both the conditions

that fed that instability and how the eddy will further evolve.

Importantly, we show that the energy of the EAC jet and the subsequent separation latitude has a strong impact on error in the representation of the EAC eddy field. A low MKE jet and southern separation will have lower representation error. This suggests that increased poleward transport of ocean heat, which is typically associated with a poleward separation latitude, should be easier to capture in models with fewer, more southerly observations.

Lastly, we suggest some sampling strategies for optimal reduction in error. For example, sampling in the high energy EAC eddy field will have a far-reaching improvement on the representation of the EAC. Likewise, sampling strategies could be designed to adapt dynamically to oceanographic conditions with higher resolution sampling upstream or downstream during northern EAC separation offset by lower resolution sampling in the downstream region during southern separation. Together





with suggested sampling strategies, the links we show between subsurface observations, ocean heat content and EAC dynamics
should help to improve predictability of the EAC and associated eddy field.

*Code and data availability.* The free-running EAC ROMS model forcing conditions are sourced from the Commonwealth Science and In-
dustrial Research Organisation (BRAN2020; available at https://research.csiro.au/bluelink/outputs/data-access/) and the Bureau of Mete-
orology (BARRA-R and ACCESS; http://www.bom.gov.au/research/projects/). Along-track SSH data is available from the E.U. Coper-
nicus Marine Service Information (https://doi.org/10.48670/moi-00146). Model configurations for the free-running and DA simula-
tions are identical to those used in previous simulations (available online at https://researchdata.edu.au/high-resolution-22-version-20;
https://researchdata.edu.au/high-resolution-reanalysis-version-2017). Output and configuration files for the simulations conducted here have
been submitted for storage to the Australian Research Data archive. The model software (ROMS v3.9) is open-source and available at
https://www.myroms.org/.

## Appendix A: Comparison against the long-term mean

In Fig. A1a-b, the spatial mean EKE and volume transport of the Reference state are compared against the mean values of
these same quantities derived from the free-running 1994–2016 simulation (e.g. Li et al., 2021). Southward volume transport
(which we calculate here as southward flow above 2000 m and west of 155° E, along a transect at ∼28° S) varies between
0 and 61 Sv (with a mean of $18 \pm 15$ Sv southwards), which compares favourably to the long-term (1994–2016) mean and
standard deviation in volume transport of $21 \pm 14$ Sv. The time series of volume transport is also shown to generally fall within
the bounds of 2 standard deviations of the long-term volume transport (Fig. A1b).

The time-mean EKE over the whole domain of the Reference state ($0.12 \pm 0.027 \text{ m}^2 \text{ s}^{-2}$ ) is very close to the long-term
mean and standard deviation in EKE of $0.12 \pm 0.028 \text{ m}^2 \text{ s}^{-2}$ . Likewise, except for several periods of rapid fluctuations, the
EKE of the Reference state also generally falls within 2 standard deviations of the long-term simulation (Fig. A1b).



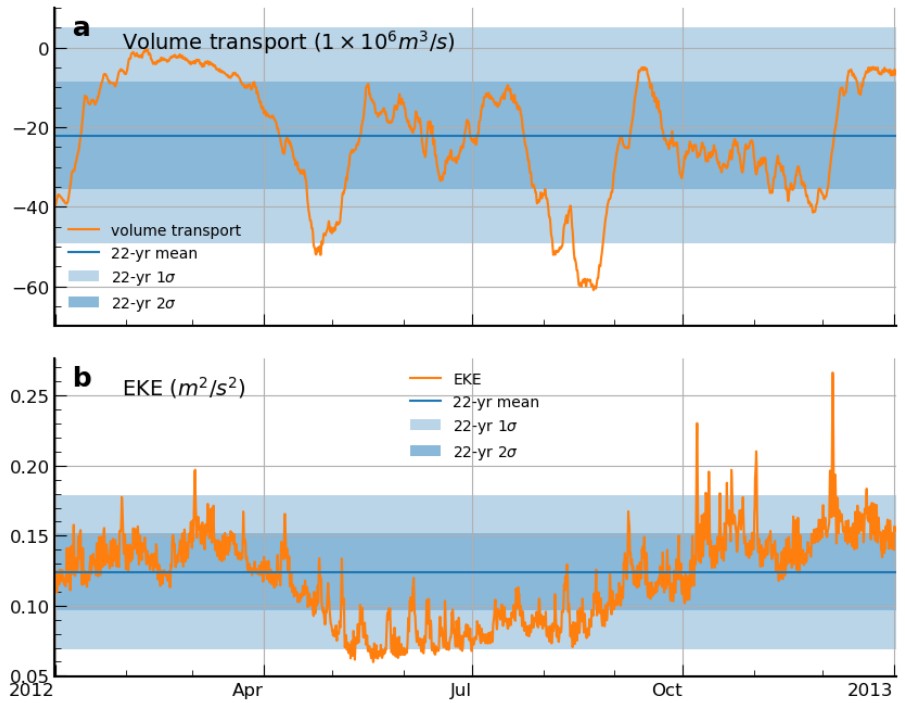

**Figure A1.** Volume transport and EKE conditions of the 1-year free-running simulation. (a) Integrated volume transport through a transect at ∼28° S (orange line). The blue line is the mean volume transport from the 1994–2016 long-running simulation, while the darker and lighter shaded regions show ±1 and ±2 standard deviations. Likewise, (b) shows the spatial-mean EKE over the whole domain (orange line). The blue line and shaded regions showing the mean, ±1 and ±2 standard deviations, respectively, in EKE for the long-running 1994–2016 simulation.

## Appendix B: Additional comparison between reference state and OSSEs

The mean SSH in the Reference state shows SSH maxima along the east coast (Fig. B1a). RMS error in SSH is relatively similar between each OSSE. All of the OSSEs assimilate the same SSH observations, so the similarities indicate that the subsurface observations have little impact on the representation of SSH (Fig. B1b-e). The region of peak RMS error matches the region of high variability in SSH (compare high RMS in Fig. B1b-e to contours in Fig. B1a).

The EAC eddy field is clearly visible in the mean EKE for the Reference state (Fig. B1f). All OSSEs exhibit similar error in

representation of this feature (Fig. B1g-j), with similar magnitudes of error to the error (standard deviation) in the EKE of the Reference state (compare high RMS in Fig. B1g-j to contours in Fig. B1f). At 500 m depth, the EKE field is weaker, though it shares the same spatial footprint as at the surface (Fig. B1k). All OSSEs perform similarly, though the inclusion of the southern XBT transect (either by itself or together with the northern transect) produces the lower RMS error (Fig. B1n-o).





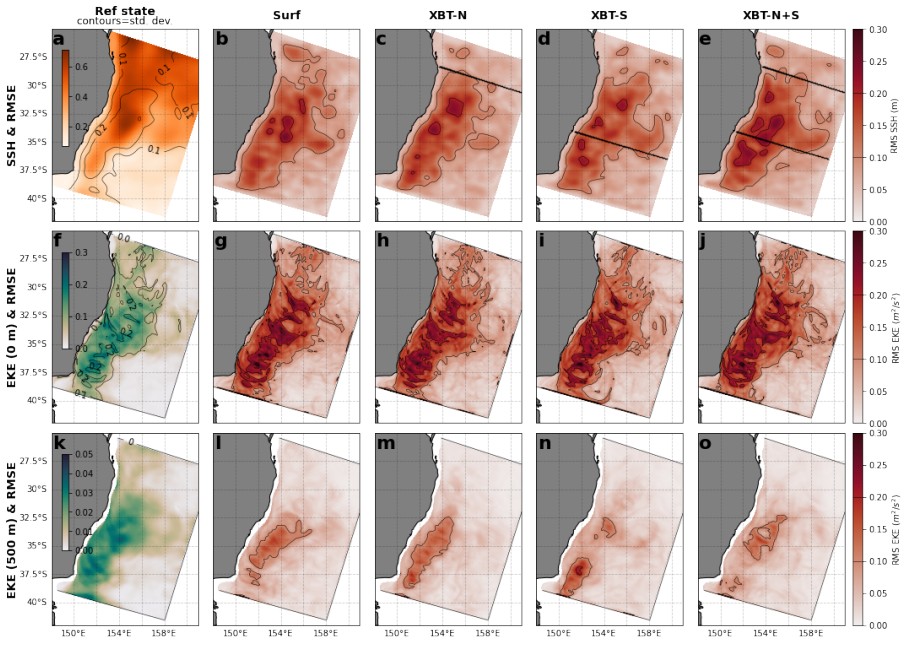

**Figure B1.** The differences in spatial representation of SSH and EKE between the Reference state and each OSSE. (a) The mean SSH of the Reference state, with contours of standard deviation in that field. The RMS error between the Reference state and the OSSEs (b) Surf, (c) XBT-N, (d) XBT-S, (E) XBT-N+S. The second row (panels f-j) shows the same diagnostics for surface EKE. The third row (panels k-o) shows the same fields for the EKE at 500 m . Note the identical colour scales for all RMS error plots. In Reference state panels, contours are the standard deviation in the respective field, while in all OSSE panels, the contours are of RMS error. Contour intervals are the same for all panels and are shown as a line in the RMS error colourbar. Black lines in panels c-e indicate location of the XBT lines.

## Appendix C:  RMS error in Upper Ocean Heat Content (UOHC)

The Root-Mean-Square Error (RMSE) in UOHC is calculated between the Reference state and each OSSE (Table C1). The RMSE values are calculated for both UOHC calculations: the upper 700 m and the upper 2000 m , and for all three regions of interest, which are designated in Fig. 3.



**Table C1.** Root-Mean-Square Error (RMSE) in Upper Ocean Heat Content (UOHC) for regions a, b, c (see Fig. 3b), over the top 700 m and top 2000 m . The RMSE value is shown for each experiment, compared to the Reference state.

| Region | Experiment name | UOHC RMSE ($\times 10^{22}$ J) | |
| --- | --- | --- | --- |
| | | 700 m | 2000 m |
| a | SSH+SST | 0.018 | 0.020 |
| | XBT-S | 0.011 | 0.013 |
| | XBT-N | 0.0090 | 0.013 |
| | XBT | 0.010 | 0.015 |
| b | SSH+SST | 0.035 | 0.027 |
| | XBT-S | 0.0093 | 0.0083 |
| | XBT-N | 0.015 | 0.0083 |
| | XBT | 0.018 | 0.012 |
| c | SSH+SST | 0.019 | 0.018 |
| | XBT-S | 0.0095 | 0.0086 |
| | XBT-N | 0.0099 | 0.011 |
| | XBT | 0.0090 | 0.012 |

*Author contributions.* DEG, CK, MR and SRK conceived and designed the experiments. DEG and CK performed the simulations. DEG analysed the data. All authors contributed to interpretation of the results. DEG prepared the paper with contributions from all co-authors.

*Competing interests.* The contact author has declared that neither they nor their co-authors have any competing interests.

*Acknowledgements.* This research and DG were supported by Australian Research Council Industry Linkage Grant LP170100498 to MR, SRK and CK. Former model development was supported by Australian Research Council grants DP140102337, LP160100162 to MR. This research was undertaken with the assistance of resources and services from the National Computational Infrastructure (NCI), under the grant *fu5*, as well as computations using the computational cluster Katana (https://doi.org/10.26190/669x-a286) supported by Research Technology
Services at UNSW Sydney.





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
