# Peer review of "Observing system simulation experiments reveal that subsurface temperature observations improve estimates of circulation and heat content in a dynamic western boundary current"

_Geoscientific Model Development, 2022_

## Author Comment (AC1)

**Response to reviewer comment RC1**

Reviewer comments are presented first in blue italics, then followed by the author's response in normal font. Line numbers are referring to the original manuscript and are denoted as L145 for Line 145.

**Major comments:**

*On the generation of the baseline: In this paper, the "baseline" state is created by re-running the model using perturbed initial conditions. However, deviations from reality do not arise only from uncertainty on initial conditions. Uncertainties on mixing parameterizations and meso-/submesoscale representation can also be important. Also, real observations always have more noise (internal waves, turbulence…). Random noise could be added to synthetic observations to increase their realism. By using a baseline generated by the same dynamical model, are you not placing your assimilation system in a very favourable situation?*

We agree that deviations from reality do not only arise from uncertainty in initial conditions. They also arise from uncertainties in boundary conditions, atmospheric forcing, parameterisations and other model physics and unresolved physical processes (e.g. internal tides). However, the purpose of an OSSE is to test how effective the data assimilation of the chosen observation platforms is at improving the model estimates of the ocean state (given a known 'Reference State'). As a result, we assimilate into a model estimate of the ocean state that has errors that are typical of a 'real' operational DA system.To achieve this, we perturb the initial conditions with an offset in time to produce errors of a realistic level in the initially perturbed ocean state. By perturbing the initial conditions as we have done, we achieve a model estimate of the ocean state that deviates considerably from the Reference State in terms of the temporal and spatial evolution of the eddy field (as discussed in lines 150-155). We then assimilate synthetic observations from the Reference State that have random noise added. The magnitude of the random noise is such that the errors are normally distributed with a standard deviation equal to the realistic observation errors, as was explained at L179-181, L192-195, L207-2011. Our OSSEs are therefore showing how the DA minimises errors that come about from incorrect initial conditions (while the forcing, boundary conditions and model physics are the same). This systematic approach provides useful results to show the impact of surface and subsurface observations on the metrics that we evaluate.

We also discussed (L150-155) that we tested a variety of initial condition perturbations, and all led to errors growing to approximately equal amounts. Therefore, our choice of perturbation within the selection we tested is appropriate and justified (L147-149).

Lastly, as explained above, we do indeed add random noise to the synthetic observations (see explanation above). See L179-181, L192-195, L207-211 for more details.

*Choice of synthetic observations: The authors have chosen to test XBT strategy but not any of the other observing systems, such as gliders, Argo, etc. Also, they chose to focus on temperature-only data, while salinity data are available on other platforms than XBT. Yet I imagine they should have an impact. Any reason for that? As it stands, this does not meet*

*the criteria of an objective assessment of the value and complementarity of different observing strategies?*

There are myriad observation platforms and strategies we could test. However, we purposely designed this study as a systematic approach by adding one datastream at a time. This approach allows us to explicitly test the value of each dataset as it is added to the system. This then explicitly shows the value of each XBT line separately and in concert, and allows testing of the upstream versus downstream impact of subsurface temperature observations. This was motivated by results of Kerry et al (2018) who showed observations of the eddy field were more impactful than observations of the upstream EAC jet, as discussed at lines 438-456.

Indeed in two earlier papers, we have assimilated glider and Argo data (see Kerry et al., 2016, and Kerry et al., 2018) to assess the impact of gliders and Argo data (including salinity) on the resulting DA simulation. Additionally, Argo and Glider data are more sparse in time and/or space, so it would not allow for a fair test of how they improve, for example, the subsurface structure of temperature. Whereas with a regularly repeated XBT line, we can robustly test this.

We have added the following sentences to the introduction to explain our reasoning for focussing on subsurface temperature observations, changing the following lines (L71-73):

> "In particular, we examine the role of surface and subsurface temperature observations in improving the simulation of prominent EAC flow features, the vertical and spatial heat and velocity distributions, and ocean heat content."

To

> "In particular, we have chosen to examine the role of SSH, SST and subsurface temperature observations in improving the simulation of prominent EAC flow features, the vertical and spatial heat and velocity distributions, and ocean heat content. Subsurface observations are systematically added in separate OSSEs to show the value of each observation platform in the absence or presence of the other subsurface observations."

*Conclusions are weakly supported by the results. The description of the beneficial impact of XBT observation seems exaggerated. In particular, it is very hard to assess the impact of different assimilated datasets as the baseline case is never presented anywhere. Another puzzling aspect is that the case assimilating all "data" is markedly less good than the one with only one XBT section.*

We have followed the reviewers request and explored the comparison against the perturbed ('Baseline') free-running simulation. The plot below shows the SST and SSH fields in the Reference state, and the RMS error for that field in the Surf and XBT OSSEs, and the Baseline.

[Figure]

As expected, all OSSEs perform considerably better than the Baseline run at the surface, as assimilation of SSH and SST observations ensure the correct spatial and temporal evolution of the eddy field at the surface. This figure will be added to the supplementary (and referred to in the new paragraph (see below). Consequently, we have removed the reference to the SSH fields in Fig. B1.

We have also updated the original Figure 5 spatial RMS plot (see below) to include a comparison with the perturbed run. This demonstrates the improvement that the DA simulations provide over a free-running simulation, and is a useful addition to the manuscript. We thank the reviewer for this suggestion.

[Figure]

Despite the improved representation of the surface eddy field, the experiment that assimilates surface only observations (Surf) has greater errors than the baseline at 250m,

500m and 1000m depth. This highlights the poor subsurface representation when assimilating only surface observations, consistent with results of Zhang et al (2010) and Zavala-Garay et al (2012). All of the OSSEs that include XBTs (XBT-N, XBT-S and XBT-N+S) provided considerable improvement on Surf and provide errors that are below the Baseline run at 250m and 500m in the eddy field region. This is because the dynamic evolution of the eddy field is better simulated with data assimilation. RMS errors at 1000m are slightly greater for all OSSEs than the errors in the Baseline run, which is explained by a) the lower natural variability at this depth (necessarily making the free-running Baseline more accurate through lower impact of that initial perturbation) b) the corrections that the 4DVar simulation makes often leads to degradation in vertical representation.

Indeed, better subsurface representation of the ocean circulation, including eddies, requires further improvements to the 4DVar system configuration (specifically, improved background error covariance estimates) and is the focus of our future work. However, the purpose of this study is to show the value of subsurface temperature observations given a carefully configured ROMS 4DVar system (Kerry et al., 2016), with a particular focus on the EAC and EAC eddy field.

We appreciate the suggestion from the reviewer to consider the Baseline, and we feel it has raised some interesting points. So, in addition to updating Figure 5 to include the Baseline RMS fields, and the new supplementary figure of a SSH and SST comparison to the Baseline, we have added the following paragraph at line 290:

"Assimilation of SSH and SST observations ensures the correct spatial and temporal evolution of the eddy field at the surface, leading to all OSSEs performing better than the Baseline (cf. Fig. B2). All of the OSSEs that include subsurface observations produce better representation of subsurface conditions at 250m and 500m than the Baseline simulation (cf. Fig 5c-e to Fig 5f; Fig 5i-k to Fig 5l). This is because the spatial and temporal evolution of the dynamic eddy field is better simulated with data assimilation. RMS errors at 1000m are slightly greater for all OSSEs than the errors in the perturbed run (cf. Fig 5n-q and Fig 5r), which is explained by: the lower natural variability at this depth (necessarily making the free-running Baseline more accurate through lower impact of that initial perturbation); and, the corrections that the 4DVar simulation makes often lead to a degradation in vertical representation. From hereon, we exclude comparison to the Baseline so as to focus on the impact of the subsurface observations compared to the Surface-only experiment."

With regards to the impact of assimilating both XBT lines, as opposed to just one XBT line, Fig. 5 and Fig. 7 give examples of when assimilating all the observations (the SSH+SST+XBT-N+S OSSE) produces an overall better solution than when only one XBT line is assimilated. Temperature RMS at 250m is improved by XBT-N+S, compared to Surf, XBT-N and to a lesser extent XBT-S (Figure 5e compared to Figure 5b-d). Likewise, the vertical temperature RMS for the XBT-N+S OSSE across all transect locations, performs competitively compared to the Surf, XBT-N or XBT-S OSSEs (Figure 7). However, there are certain metrics for which this is not the case. To highlight this, we have modified the following line at L323:

"The XBT-N+S OSSE has relatively good representation of temperature along this transect and much improved representation below 1000 m (Fig. 7o)."

To:

"The XBT-N+S OSSE has relatively good representation of temperature along this transect and much improved representation below 1000 m (Fig. 7o); and indeed considering all transects (Fig. 7e,j,o), the XBT-N+S OSSE has low RMS error, as opposed to the single XBT OSSEs, which have high RMS at some transect locations (e.g. Fig. 7h,n)."

This is explained by the least squares best fit for both sets of observations being one with a degraded fit at both locations, as we discuss at L377-379. We have also added some further discussion of this at L422.

There are other examples in the literature where the addition of observations degrades estimates. For example, Zhang et al. (2010) showed that assimilating surface current observations degraded subsurface temperature estimates, and Siripitana et al. (2020) showed that assimilation of subsurface observations is often at the expense of the fit to surface observations.

*Overall, I am not convinced this paper is best suited for publication in Geophysical Model Development. There is no model development presented here.*

As per the journal scope, GMD does not only contain papers focussing on model development. This manuscript is submitted as an "model experiment description paper". For the manuscript type, the GMD instructions for manuscript type 4, (https://www.geoscientific-model-development.net/about/manuscript_types.html#item4) note that "Configurations and overview results of individual models can also be included as well as descriptions of the methodology of experimental procedures such as ensemble generation. Such papers should include the discussion of why particular choices were made in the experiment design and sample model output." Our manuscript is an overview of the results of a model experiment, and fits the criteria set out in the GMD manuscript types guidelines.

Minor comments:

1. *l. 1 and l. 16: WBCs are not necessarily poleward flowing. Maybe add here that you focus on Subtropical WBCs.*

We have modified L16 as suggested to:

"Subtropical Western Boundary Currents (WBCs) transport warm and saline towards the poles"...

The mention of poleward flow in the abstract is: "Western boundary currents (WBCs), form the narrow, fast-flowing poleward return flows of the great subtropical ocean gyres", which is ostensibly correct.

2. *l. 23: Not all DA methods combine model-obs in a "dynamically consistent way". State here that you are using 4DVar with 5 days windows.*

The text has been modified as follows:

Data Assimilation (DA) combines observations and a numerical model in a dynamically consistent way such that the result is a better estimate of the ocean circulation than either alone (Moore et al, 2019).

To:

Data Assimilation (DA) is the combination of observations and a numerical model, such that the result is an optimal estimate of the ocean state (Moore et al., 2019).

We state at L81 and L97 that we use 4DVar with a 5 day window.

3. *l. 86: why is the resolution varying so much in such a small domain? And how?*

The model resolution varies between 2.5 km on the continental shelf and slope to 6 km in the deep ocean over a distance of > 1500km, with a linear transition between. The gradual change in resolution is shown in the below plots:

[Figure]

These plots show the variation in horizontal resolution (in metres), with a gradual increase from 2500m to 6000m, across the domain in the across-shelf direction (xi_rho coordinate) of the model.

We have modified the manuscript to say:

"Horizontal resolution is 2.5km over the continental shelf and slope, and increasing linearly to 6km in the deep ocean"

There are also many previous studies that have used this model domain successfully (e.g. Kerry et al., 2016; 2018; Kerry and Roughan, 2020; Li et al., 2020; Li et al., 2021).

4. *l. 255: These results are very confusing. How is it possible that the worse experiment be the one assimilating the largest amount of data? Why do you not show the baseline case to see how bad the run is without assimilation? Looking at this figure only would actually lead me to conclude that XBT data have a very minor impact if not a detrimental one on the representation of the circulation.*

It has been widely shown that increasing the number of observations that the assimilated analysis has fit to will lead to a reduced fit to any one single observation. For example, Siripatana et al. (2020) found that including extra datastreams (HF radar currents and moorings) in addition to traditional observations (e.g. satellite SSH and SSH observations) degraded the model representation of SSH and SST. Zhang et al. (2010) showed that HF radar observations of currents degraded the subsurface temperature forecast, which they attributed to a lack of cross-variable covariance estimates.

We have demonstrated that subsurface observations improve estimates of key quantities (e.g. temperature at depth), which is especially noticeable in the high EKE region. It has also been widely shown that the assimilation of subsurface observations has a strong impact on ocean state estimates. For example, Moore et al. (2011) presented evidence for the considerable impact that subsurface observations have relative to surface observations. Zavala-Garay et al. (2012) also showed that surface-only observations result in poor estimates of the true subsurface ocean, and while this is improved dramatically in the presence of XBT observations, away from those observations, the improvement in subsurface skill was poor.

We have added the below paragraph to the discussion section at L421:

"We have demonstrated that subsurface observations improve estimates of key quantities (e.g. temperature at depth), which is especially noticeable in the high EKE region. It has also been widely shown that the assimilation of subsurface observations has a strong impact on ocean state estimates (Moore et al., 2011; Zavala-Garay et al., 2012). However, increasing the number of observations can lead to a reduced fit in any one single observation (e.g. compare RMS in XBT-N+S to XBT-S or XBT-N). This has been also demonstrated by others. For example, Siripatana et al. (2020) found that including extra datastreams (HF radar currents and moorings) in addition to traditional observations (e.g. satellite SSH and SSH observations) degraded the model representation of SSH and SST. Zhang et al. (2010) showed that HF radar observations of currents degraded the subsurface temperature forecast, which they attributed to a lack of cross-variable covariance estimates."

We have addressed the comment on the comparison with the baseline in an above reply.

5. *I. 389: I do not understand which feature in the deep MKE is better represented thanks to XBT assimilation. On the contrary, it seems to me that deep properties are somehow worsened by XBT assimilation when compared with the Ref.*

We apologise for not being clear, we have replaced the discussion with the following line:

"The better reproduction of key EAC features in the surface MKE fields (e.g. more energy in the southern extension, return flow and eastern extensions; Fig. 4) when subsurface temperature observations (together with surface observations) are assimilated must be due to the improved subsurface conditions impacting the surface circulation."

6. *Discussion is mostly a repetition of the Result section. What is the purpose?*

We have 5 sections in the discussion, each with their own sub headings (see Lines 382, 422, 437, 457, 492) each with a different theme. In this section we place our results in context of the literature and show how the different OSSEs resolve the dynamics of the EAC system and how the different experiments contribute to resolving upper ocean heat content in the system. The final section in the discussion is where we discuss the results in the context of observing system design - which is an important contribution to the ocean prediction and ocean observing community.

---

## Author Comment (AC2)

**Response to reviewer comment RC2**

Reviewer comments are presented first in *blue italics*, then followed by the author's response in normal font. Line numbers are referring to the original manuscript and are denoted as L145 for Line 145.

**Major comments:**

*In general, if much observations are assimilated, the result will show higher accuracy. However, the authors have shown that the experiment assimilating both southern and northern XBT observations did not show the best performance. I am wondering if the cost function was properly reduced. How were the convergence conditions defined, such as the ratio of the final to the initial value of the cost function or the gradient of the cost function?*

We chose to define a set number of inner loops to reduce the cost function. This was based on previous work of Kerry et al., (2016), who showed that 15 inner loops achieved an acceptable reduction with a reasonable computational cost, and is common in 4D-Var applications. The number of inner loops we used (15) shows similar convergence between all the OSSEs, with a likewise similar final nonlinear model cost function ratio (crosses). We show this in the below plot.

[Figure]

The ratio of the final observation cost function to the initial observation cost function shows minimal differences between the Surf, XBT-N, XBT-S and XBT-N+S OSSEs.

[Figure]

This shows that all OSSEs have similar cost function reduction, with the mean values of the ratio of final to initial total cost function ranging between 0.73 and 0.80 and the time-mean ratio of final to initial observation cost function ranging between 0.77 to 0.78. That is, all OSSEs have similarly reduced cost functions. Please refer to Eq 6 in Kerry et al, 2016) for more details of the cost function.

We now include in the text the following sentence at L98:
"We chose 15 inner loops to reduce the cost function, based on previous work of Kerry et al., (2016), who showed that this number of inner loops achieved an acceptable reduction with a reasonable computational cost. Similar cost function reduction is achieved for all OSSEs, with a time-mean ratio of final to initial cost function ranging between 0.73 to 0.80."

And following from that, we added the following sentences to elaborate on the quality control methods we employed:
"Background quality control was applied to eliminate observations that are poorly represented, following the method described in Moore et al (2013). Only observations that satisfy $d_i^2 < \alpha^2(\sigma_b^2 + \sigma_o^2)$ are assimilated, where $d_i$ is the innovation, the quality control parameter $\alpha=4$ and $\sigma_b$ and $\sigma_o$ represent the prior background and observation errors. In all OSSEs, about 20% of SST observations were rejected by this criteria while all of the SSH observations were assimilated. For the subsurface XBT observations, all observations were used for the single transect OSSEs, while in XBT-N+S between 20-40% of observations were rejected, due to the innovation being too large. This method has been applied in several other recent 4D-Var studies (Levin et al., 2021; He et al., 2022)."

*The authors also show the XBT-S experiment is the best among the OSSEs. How does the observed information propagate upstream (to the north)? As I see the Figure 6 in Kerry et al. (2018), the impact of the XBT is limited to the vicinity of the observation latitude. An influence scale of 600 km seems distant relative to the spatial scale of analysis increment. If there is an impact of the XBT-S observations on the northern boundary, it should be clarified in the manuscript.*

There may be some misunderstanding of Figure 6 of Kerry et al,. (2018): In that figure, the XBT observations taken between 34S to 36S have an impact on transport at all cross sections between 27.5S and 36.2S. Further, although the total impact is low, the impact per observation (bottom panels) is high.

We are unsure what the reviewer means by "An influence scale of 600 km seems distant relative to the spatial scale of analysis increment." Analysis increments are applied to the boundary, surface forcings, and initial conditions, with the spatial scales controlled by the background error covariances as described in Kerry et al. 2016. The use of the adjoint and tangent linear models in 4D-Var introduce flow dependence to the covariances.

It is true that the information provided by the observations cannot propagate 600km upstream by advection over the 5 day windows, but in 4D-Var, information can propagate via a number of physical processes as discussed in Kerry et al, (2018). In that study, observation impact is shown to be far reaching both up and downstream. There are other

examples of far-reaching observation impact in the literature, for example in Siripitana et al (2020) and Powell et al., (2017).

*These points are discussed in section 4, but it would be better if you could briefly introduce the discussion part in the results section, for example after L316. When I read section 3.4, I wondered how the southern XBT line would affect the upstream regions far from 600 km.*

In this paper we have tried to mostly limit the results section to focussing on results, and the discussion section to the discussion of the key ideas emerging from the results. However, we take the reviewers point, and have added at L316:
"The mechanisms that might lead to such action at a distance are discussed further in Section 4.1 and 4.3."

Likewise, we have added further discussion of the results showing the reduced impact from XBT-N+S at L323:

"The XBT-N+S OSSE has good representation of temperature along this transect and much improved representation below 1000m (Fig.7o; depth-averaged RMS of 0.6C); and indeed considering all transects (Fig.7e,j,o), the XBT-N+S OSSE has low RMS error, as opposed to the single XBT OSSEs, which have high RMS at some transect locations (e.g. Fig.7h,n).

And more at L422:

"We have demonstrated that subsurface observations improve estimates of key quantities (e.g. temperature at depth), which is especially noticeable in the high EKE region. It has also been widely shown that the assimilation of subsurface observations has a strong impact on ocean state estimates (Moore et al., 2011; Zavala-Garay et al., 2012). However, increasing the number of observations can lead to a reduced fit in any one single observation (e.g. compare RMS in XBT-N+S to XBT-S or XBT-N). This has been also demonstrated by others. For example, Siripatana et al. (2020) found that including extra datastreams (HF radar currents and moorings) in addition to traditional observations (e.g. satellite SSH and SSH observations) degraded the model representation of SSH and SST. Zhang et al. (2010) showed that HF radar observations of currents degraded the subsurface temperature forecast, which they attributed to a lack of cross-variable covariance estimates."

The last two additions were in response to RC1, and we refer to that answer for further details.

*This study focused on the XBT observation network. However, in realistic situations, other networks exist, such as Argo floats, HF radars, and sea gliders. Why did the authors investigate the impact without using other observation network? Of course, I admit that it makes sense to evaluate the impact of XBT observations.*

As RC#1 also made this remark, we answer similarly: There are many observation platforms and strategies we could test. However, we purposely designed this study as a systematic approach by adding one datastream at a time. This approach allows us to explicitly test the value of each dataset as it is added to the system. This then explicitly shows the value of

each XBT line separately and in concert, and allows testing of the upstream versus downstream impact of subsurface temperature observations.

Indeed in two earlier papers, we have assimilated glider and Argo data (see Kerry et al., 2016, and Kerry et al., 2018) to assess the impact of gliders and Argo data (including salinity) on the resulting DA simulation. Additionally, Argo and Glider data are more sparse in time and/or space, so it would not allow for a fair test of how they improve, for example, the subsurface structure of temperature. Whereas with a regularly repeated XBT line, we can robustly test this.

Following RC#1 and this review, we have added the following sentences to the introduction to explain our reasoning for focussing on subsurface temperature observations, changing the following lines (L71-73):

> "In particular, we examine the role of surface and subsurface temperature observations in improving the simulation of prominent EAC flow features, the vertical and spatial heat and velocity distributions, and ocean heat content."

To

> "In particular, we have chosen to examine the role of SSH, SST and subsurface temperature observations in improving the simulation of prominent EAC flow features, the vertical and spatial heat and velocity distributions, and ocean heat content. Subsurface observations are systematically added in separate OSSEs to show the value of each observation platform in the absence or presence of the other subsurface observations."

*Temperature observations per model cell may be too dense to account for representation errors. It may be better to consider the super-observation or thin out the observation. Also, if you want to show the advantage of the high density XBT observations, it would be useful to demonstrate an additional experiment which assimilates the XBT at regular observation interval. In addition, the manuscript mentions observation errors of XBT, but it is unclear whether the representation errors are considered. How about you clarify this point?*

We only apply a single observation per cell. Further, as the synthetic observations come from the same model grid, there is no need to generate super-observations as only one data point per cell is extracted.

We agree that exploring the impact of the XBT transect density would be useful and interesting. However, to properly explore this, it would require more model runs and enough additional analysis that it would broaden the scope of this manuscript too much.

Representation errors result from the discretisation of the model grid and unresolved physical processes (e.g. inertial tides). However, none of these sources are relevant to us as we sample the observations from a free-running model that has the same model grid and resolves the same physical processes. The errors we apply add noise such that the observations are realistic of real observations (where there would be representation errors).

We have added the modified sentence to clarify this point at L185, from:
"As a result, we choose each model point as an observation location."
To
"As a result, we choose each model point as an observation location, and have no need to superpose or thin observations. Further, and likewise for the other synthetic observation types, there are no representation errors, as we sample observations from the same model; realistic errors are added with random noise (see below)."

Also note that we compare the realistic XBT transect temporal and spatial resolution (L199) to our synthetic XBT transect temporal and spatial resolution (L200 & L203).

*The authors often use the term "best" to specify which experiment represents good performance (such as L262 and L356). However, it should be better to quantify the experiment using statistical metrics such as area-averaged biases and RMSEs. In addition, it would be useful to specify the improved ratio of each OSSE relative to the surf-only experiment or the baseline when comparing the impact of observations across each OSSE (for example L297 and Figure 6).*

In light of the reviewer's suggestion, we have added the area-averaged RMS error value to the corner of each panel in Figures 5, 6, 7, 10 and the new Supplementary Figure B2. Because the eddy region (box b, as shown in Figure 3b) is the region of highest RMS and is of particular interest for representing eddy dynamics, we have also included the mean RMS for this region, and both values are noted in each panel. We have updated the captions accordingly.

We have adjusted the text to include better quantification with these RMS values:

At L274:
"For each field and OSSE, the area-mean RMS value, as well as the mean RMS error for the high EKE region (box b; Fig.3b) are shown in Fig.5."

At L278:
"The presence of the subsurface observations greatly improves 250m temperature RMS (Fig.5c-e; compare RMSE values in the eddy region of 1.6C - 1.8C to 2.5C for Surf)."

L282:
"The addition of the north and south transects in the XBT-N+S OSSE (Fig.5e) improves temperature RMS compared to the Surf OSSE, and has a similar spatial pattern in RMS error to the southern XBT OSSE, with a mean RMSE of 1.7C in the eddy region, compared to 2.5C for the Surf OSSE."

L285:
"The OSSEs with either a northern or southern transect of XBT observations (Fig.5i-j) display relatively low RMS error, especially compared to the surface only observations (Fig.5h; area mean RMS of 0.9C compared to 1.7C for Surf)."

We have added a line to the discussion on the Baseline simulation (see RC#1 comments):

"The area mean RMS values for the Baseline are marginally lower than XBT-S for temperature at 250m and all OSSEs at 500m; when considering the mean over the eddy region, the Baseline simulation has a higher mean RMS than the subsurface OSSEs. This is because the spatial and temporal evolution of the dynamic eddy field is better captured with data assimilation."

L319:
"The addition of subsurface observations in either the north or the south reduces RMS error near the separation zone (Fig.7h-i; depth averaged RMS values of 0.7C, as compared to 1.0C for the Surf OSSE), indicating an improvement in the representation of heat content carried by eddies."

L321:
"The XBT-N+S OSSE has good representation of temperature along this transect and much improved representation below 1000m (Fig.7o; depth-averaged RMS of 0.6C)..."

L368:
"...however observations in this upstream region (Fig.10b,d; XBT-N and XBT-N+S) lead to the largest improvement in error in representation of the EAC jet, with depth-averaged RMS values of between 0.08m/s -- 0.09m/s, as compared to 0.1m/s for the Surf and XBT-S OSSEs."

L372:
"During northern separation at ~32S (Fig.10i), both experiments with a single subsurface XBT transect (XBT-N and XBT-S, Fig.10j-k) have the lowest RMS error (depth averaged RMS of 0.14m/s compared to 0.19m/s for XBT-N+S and 0.25m/s for Surf),..."

L376:
"...velocity representation along the ~32S transect is better in the presence of southern rather than northern subsurface observations (compare Fig.10o to Fig.10n; RMS error of 0.08m/s compared to 0.13m/s)."

With regards to the reviewer's second point, we originally calculated the ratio of improvement, defined as the percentage ratio of the RMS of the XBT-N, XBT-S and XBT-N+S OSSEs, compared to the RMS of the Surf OSSE. However, we went back to pure RMS values, because it was easier to interpret the figures and associated particular patterns in the plots with physical mechanisms, for example, increased RMS in the Surf OSSE in the subsurface velocities near to the return flow.

*The upper ocean heat content (OHC) in the OSSEs in Figure 8a and 8b showed similar temporal evolution until March or April 2012. Why the observation impacts were less? Conversely, why did the difference of the upper OHC occur after April 2012? Does it relate to the EAC phase?*

We thank the reviewer for this comment. It seems likely that the patterns described do relate to periods of southern and northern EAC separation. For example, the OSSEs do show similar evolution in UOHC until around March, before they diverge in their temporal evolution. This coincides with a period of southern separation (until an eddy shedding event

in mid March). Conversely, in late September, the EAC separation region moves northwards, which coincides with a period of increased difference between the OSSE predicted UOHC and the Ref state (e.g. see Figure 8a).

We have added a new paragraph to highlight this, at L342:
"The periods of increased or decreased error between the Ref state and the OSSEs, as well as between each OSSEs, generally coincide with periods of more northerly or southerly EAC separation latitude (See Section 3.6 and Section 4.4). For example, during mid January and May 2012 there was relatively good agreement in UOHC, which coincided with periods of southern EAC separation. Conversely, in October, when the EAC separated further north, most OSSEs had increased error compared to the Ref state. Interestingly, while all OSSEs have generally worse representation of UOHC in the upstream box a region, there are periods of lower RMS, for example in early to mid August, when there is a coherent EAC jet through the box a region, and likewise, with box c in late April. This suggests that the ability to represent UOHC depends on the location of the coherent EAC jet and separation latitude."

This now nicely segues into the next section on separation latitude. We have added an additional note at L471:
"The error in UOHC between the OSSEs and the Ref state is also related to separation latitude, as discussed above (Section 3.5)."

*The vertical temperature section for the ~35.5S transect in Fig. 7n shows a large RMSE below 1000 m depth, whereas the upper OHC above 2000 m of the XBT-S in Fig. 8b and 8c (green dash line) is better represented than in other experiments. Based on the fact that the 35.5S transect locates between box b and box c shown in Figure 3b, the vertical temperature in Figure 7n would be expected to be best represented. However, it is not, and appears to be inconsistent. How should we interpret this point?*

This apparent inconsistency is because the UOHC in box c is further south than the transect location of 35.5S.

To demonstrate this, we present the transect through the middle of box c, as demonstrated here (red line is at ~36.6S):

[Figure]

The mean RMS in temperature through this transect is shown in the below plot:

[Figure]

Now we see that the RMS error is lower at depth in XBT-S (compare panels s to n), as well as lower RMS in XBT-N & XBT-S compared to Surf. This matches what we see in the UOHC time series plot, demonstrating there is no inconsistency.

**Specific comments**

*P5 L134: The Reference state and the baseline experiment is forced by BARRA-R while the OSSEs have applied the ACCESS reanalysis as surface forcing. Is it right? If so, how about you emphasize that such condition leads to additional perturbations?*

That is correct. We have added the following sentence to L155:
"Note that the different surface forcing conditions between the free run (BARRA-R) and the OSSEs (ACCESS) leads to an additional source of error that the DA system must reduce."

*P6 L143-149: Was the initial condition for OSSE chosen from the Reference state 8 days later or 8 days earlier as perturbation? In other words, is it a lagged initial condition? It would be easier to understand if you describe the date of the initial condition used in OSSE.*

We have modified this line as follows:
"We initialised each OSSE with initial conditions that were 8-days offset from those that were used to initialise the Reference state."
To:
"We initialised each OSSE with initial conditions that were 8-days offset from those that were used to initialise the Reference state (i.e. begin the OSSE at 02 December 2011 with conditions from 10 December 2011)."

*P9 L198: Transect PX30 represents the Brisbane to Noumea line (southeast direction) in this study. However, in the previous studies (Kerry et al., 2016, 2018), PX30 refers to the Brisbane - Fiji route (northeast direction). Does this mean the observation line PX30 has changed?*

We thank the reviewer for pointing this out. As seen on the NOAA website, the PX30 line goes from Brisbane to Noumea or Fiji (http://www-hrx.ucsd.edu/px31.html). We also make clear that our XBT transects are chosen to approximately represent these ship tracks: "XBT locations were generated to very approximately match XBT deployments.." at L197.

*P20 L398: It should be better mention about the EKE based on the Figure B.*

We have added a reference to the figure at the suggested location, and in the following sentence:
"Surface EKE is not further improved with the assimilation of temperature observations at depth (see Fig.B2a-e). Furthermore, the improvement from temperature (XBT) observations on EKE at depth (500m) is also minimal (Fig.B2g-j)."

**Technical comments**

*P4 L82: Please add "variational" for the abbreviation of "4DVar".*

Thank you - added.

*P11 Figure4: It would be useful to point out the velocity directions for the EAC southern extension, the EAC return flow and so on. It is also in Figure. 5 and 9. This study will attract not only the oceanographers but also data assimilation community. It will help for the readers who are not familiar with the EAC circulation.*

We have updated Figure 4a to include labels to the EAC Southern Extension, Return flow, EAC Eastern Extension, as shown below:

[Figure]

The caption has been updated to reflect this, as well as to point out the flow directions of these long-term features:

"Key circulation features that emerge in the long-term mean are shown, including the EAC eastern and southern extensions and the return flow, which flow eastwards, southwards and northwards, respectively."

*P16 Figure 7: Please draw the temperature color-bar for the Ref state in an easily recognizable location.*

We have moved the temperature colour bar as suggested.

**Other changes**

*Data repositories: Following the comment from the Chief Editor (see CEC#1), we updated our data and code availability statement (see CEC#1 reply):*

"The source code, forcing conditions, configuration files and output for the simulations conducted here are available at https://doi.org/10.26190/unsworks/24146. The free-running EAC ROMS model forcing conditions are sourced from the Commonwealth Science and Industrial Research Organisation (BRAN2020; available at https://research.csiro.au/bluelink/outputs/data-access/) and the Bureau of Meteorology (BARRA-R and ACCESS; http://www.bom.gov.au/research/projects/). Along-track SSH data is available from the E.U. Copernicus Marine Service Information (https://doi.org/10.48670/moi-00146). Model configurations for the free-running and DA

simulations are identical to those used in previous simulations (available online at https://doi.org/10.26190/TT1Q-NP46; https://doi.org/10.26190/5ebe1f389dd87). The model source code is open-source and available from https://www.myroms.org/."

*We have added several sentences at L493 to mention the use of the Surf-only DA setup for SAR:*

"Firstly, the presence of surface observations (SSH and SST) significantly improves the representation of surface currents, which is most important for navigation and search-and-rescue applications. "

*At L91, added "are" between "conditions" and "given".*

*Figure 3 caption:  removed "with with".*

*At L170, replaced "mission" with "missions".*

*At L276-277, added "(compare to the Baseline; Fig.5f)" to make clear that Surf degrades representation compared to the baseline.*

*At L292, replaced "of a subsurface" with "of subsurface".*

*At L328, replaced "is shown" with "are shown".*

*At L473, replaced "separation is" with "separation are".*

*At L479, replaced "phase is" with "phase are".*

*At L486, added "in recent decades".*

---

## Author Response (AR2)

**Response to reviewer comment RC2 #2**

Reviewer comments are presented first in *blue italics*, then followed by the author's response in normal font. Line numbers are referring to the original manuscript and are denoted as L145 for Line 145.

*Thanks to the authors sincerely response about my comments. It would be possible to be published after several minor comments listed below are corrected.*

*L74; sea surface height (SSH), sea surface temperature (SST)*
Have added abbreviations here and removed later definitions..

*L183: XBT is used in L110*
Have added abbreviation at L75, removed later definition of XBT.

*L298: "but by 1000 m depth the error is reduced". Is it right?*
Have modified it to say "but by 1000m depth the error is lower (as it is for all OSSEs)."

*L317: delete ":"*
We have rewritten the sentence: "This is explained by the lower natural variability at this depth, which necessarily makes the free-running Baseline more accurate through lower impact of that initial perturbation. Furthermore, the corrections that the 4DVar scheme makes often lead to a degradation in vertical representation."

*L453-454: It seems to be a similar expression by "while" and "though",*
We have removed the 'though', which has made this sentence clearer: "While there is some improvement in EKE from the XBT-S transect, there was degradation in the far south of the domain."

*L590: "surface observations struggle to represent these features at depth" This sentence should be mentioned about the impact of subsurface observations to logically connect to the meaning of the sentence "hence subsurface observations …".*
Thank you for picking this up. We have changed this sentence to: "While assimilating surface observations is effective at improving representation of key EAC circulation features at the surface, such as the return flow and the southern and eastern extensions, surface observations struggle to represent these features at depth. In this study, we found that assimilating subsurface observations is critical for improving representation at depth."